# PILOTRL: TRAINING LANGUAGE MODEL AGENTS VIA GLOBAL PLANNING-GUIDED PROGRESSIVE REINFORCEMENT LEARNING

## ABSTRACT

Large Language Models (LLMs) have shown remarkable advances in the ability to tackle agent-oriented tasks. Despite their potential, existing work faces challenges when deploying LLMs in agent-based environments. The widely adopted agent paradigm ReAct centers on the integration of single-step reasoning with immediate action execution, which limits its effectiveness in complex tasks that require long-term strategic planning. Furthermore, the coordination between the planner and executor during problem-solving is also a critical factor to consider in agent design. Additionally, current approaches predominantly rely on supervised fine-tuning, which often leads models to memorize established task completion trajectories, thereby restricting their generalization ability when confronted with novel problem contexts. To address these challenges, we introduce an adaptive global plan-based agent paradigm **AdaPlan**, aiming to synergize high-level explicit guidance with execution to support effective long-horizon decision-making. Based on the proposed paradigm, we further put forward **PilotRL**, a global planning-guided training framework for LLM agents driven by progressive reinforcement learning. We first develop the model's ability to follow explicit guidance from global plans when addressing agent tasks. Subsequently, on the basis of this foundation, we focus on optimizing the quality of generated plans. Finally, we conduct joint optimization of the model's planning and execution coordination. Extensive experiments indicate that **PilotRL** could achieve state-of-the-art performances, with LLaMA3.1-8B-Instruct + PilotRL surpassing closed-sourced GPT-4o by 3.60%, while showing a more substantial gain of 55.78% comparing to GPT-4o-mini at a comparable parameter scale.

## 1 INTRODUCTION

An *agent* can be defined as an entity capable of perceiving its environment, making decisions, and executing actions in pursuit of predefined or adaptive goals (Wooldridge & Jennings, 1995; Maes, 1995; Jennings et al., 1998). The state-of-the-art Large Language Models (LLMs), such as GPT-4 (Achiam et al., 2023) and Gemini (Team et al., 2023), have exhibited strong agent capabilities, including instruction following, reasoning, and programming, which inspires widespread efforts to develop autonomous agent systems with LLMs serving as central cognitive controllers (Song et al., 2023; Sumers et al., 2023). Nevertheless, considering the high financial costs and safety risks of close-sourced proprietary models (Li et al., 2023; Yuan et al., 2023), recent efforts have been shifted to improve such agent capabilities in open-sourced models as effective alternatives (Chen et al., 2024; Song et al., 2024; Fu et al., 2025).

Despite their potential, existing works face some limitations: *(C1) Limited Contextual Awareness of ReAct*: While the ReAct paradigm (Yao et al., 2023) is a general foundation of modern agentic systems, it lacks insight into the overarching context. The reasoning component (generated as "thought") focuses purely on immediate action, which limits its effectiveness in complex tasks requiring sequential execution. *(C2) Insufficient Coordination between Planning and Executing*: Although recent studies have incorporated planning into agent-based problem-solving process (Erdogan et al., 2025; Xiong et al., 2025), they design the planner and executor in isolation, leading

to potential mismatches between the two components. As a result, the plans generated by the planner may not be effectively followed by the executor, undermining overall task performance. *(C3)* ***Deficient Generalization of SFT***: Extensive research has been devoted to enhancing the agent capabilities of models through supervised fine-tuning (SFT) (Deng et al., 2023; Zeng et al., 2024). However, recent studies indicate that SFT tends to lead models to memorize task-specific heuristics rather than acquiring generalizable capabilities applicable to new scenarios (Chu et al., 2025).

To address these challenges, we introduce ***PilotRL***, which is a global plan-driven reinforcement learning framework for the training of LLM agents. For ***C1*** and ***C2***, we propose the adaptive global plan-based paradigm *AdaPlan* to guide the agent through complex tasks as a pilot, where global plans are dynamically generated and continuously updated throughout the execution process. The global planner and executor are implemented within a unified model to enhance their coordination and mutual adaptability. For ***C3***, we employ reinforcement learning (RL) for its high effectiveness at enhancing generalizable knowledge in LLMs (Jaech et al., 2024; Guo et al., 2025; Team et al., 2025), the training process of which can be divided into three stages: **(1) Stage 1: Executor Enhancement.** We begin by developing the executor's instruction adhesion to the global plan when addressing agent tasks. **(2) Stage 2: Global Planner Culti-**

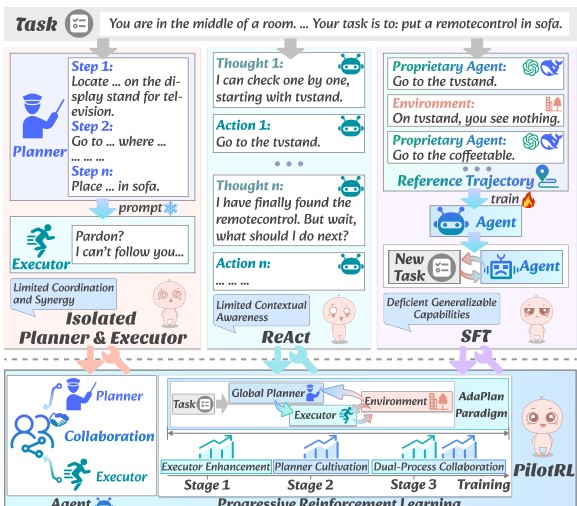

Figure 1: Comparison of PilotRL (*bottom*) with existing methods (*top*) for agent task completion.

**vation.** Building upon the global plan following capabilities acquired in Stage 1, we subsequently optimize the global planner to improve the quality of generated plans. **(3) Stage 3: Joint Optimization.** Finally, we refine the coordination between the global planning and execution of models to enhance their collaborative performance in agent scenarios.

**Contributions.** The main contributions can be summarized as follows:

- *Paradigm Innovation.* We introduce an adaptive global plan-based agent paradigm, AdaPlan, to synergize high-level reasoning with executing for long-horizon decision-making. By integrating both the global planner and executor in a unified model, our approach enables more effective coordination and improved end-to-end performance.

- *Training Framework Advancement.* Based on AdaPlan, we propose PilotRL, a global planning-guided progressive reinforcement learning framework designed for enhancing the agent capabilities of models via a three-stage process.

- *Performance and Effectiveness.* Extensive experiments indicate the superiority of PilotRL. Notably, models trained with PilotRL even surpasses closed-sourced proprietary models for agent tasks, achieving average improvements over GPT-4o and GPT-4o-mini by 2.35% and 53.90%.

## 2 PILOTRL

Assuming the scenario where an agent interacts with an environment for task solving, we present a detailed overview of our proposed PilotRL framework in this section.

### 2.1 ADAPLAN: ADAPTIVE GLOBAL PLANNING

While ReAct (Yao et al., 2023) is effective in many interactive agent tasks, its reliance on single-step reasoning and immediate action generation limits its capability in scenarios that require extended planning and coherent decision-making. To address this, we introduce the ***AdaPlan*** paradigm, which focuses on the adaptively generated and refined global plan throughout the task-solving process.

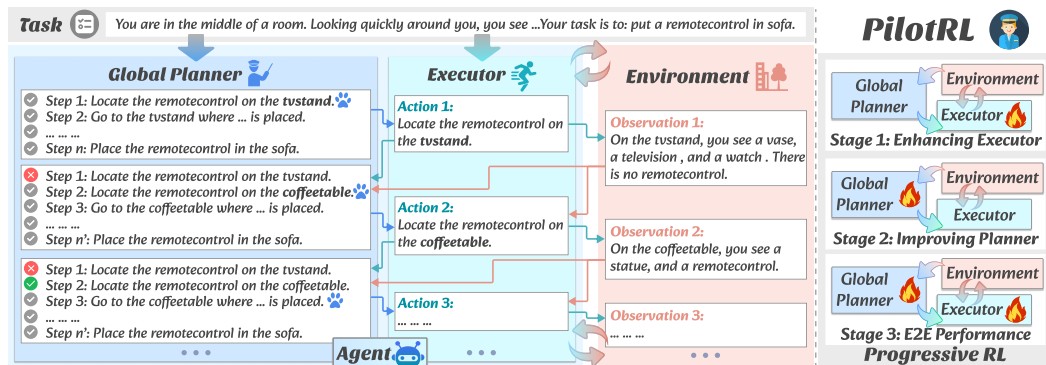

Figure 2: Overview of **PilotRL**. (*Left*) In **AdaPlan** paradigm, the *global planner* begins by processing the task instruction and generates an initial high-level plan for guidance, which is then passed to the *executor* for action generation. The observation from the *environment* is then fed back to both the *executor* for subsequent action generation and to the *global planner* for plan adaptation in response to changes or unexpected outcomes. (*Right*) The three-stage training process of **progressive RL**.

As shown in the left part of Figure 2, the agent architecture consists of two key components: the *global planner* and the *executor*. For a given task instruction $G$ and the initial context $\mathcal{C}^{(0)}$, the global planner first generates the global plan $\mathcal{P}^{(0)} = [p_1^{(0)}, p_2^{(0)}, ..., p_{N_0}^{(0)}]$ consisting of $N_0$ steps, where $p_i^{(0)}$ represents the recommended action for the executor at step $i$ under the current planning strategy. At each time step $t$ ($t \geq 1$), the executor takes an action $a^{(t)} \in \mathcal{A}$ based on: *(1)* the previous accumulated context $\mathcal{C}^{(t-1)} = \{(a^{(j)}, o^{(j)})\}_{j=0}^{t-1}$, where $o \in \mathcal{O}$ refers to observation from the environment, and *(2)* the guidance from the current global plan $\mathcal{P}^{(t-1)}$. Subsequently, it receives the resulting observation $o^{(t)} \in \mathcal{O}$, and the current turn of agent-environment interaction $(a^{(t)}, o^{(t)})$ is incorporated into the accumulated context $\mathcal{C}^{(t)}$ of the execution step $t$. The global plan $\mathcal{P}^{(t-1)}$ is then iteratively refined according to the task goal $G$ and the accumulated context $\mathcal{C}^{(t)}$ to facilitate the next execution, resulting in $\mathcal{P}^{(t)}$. Each $p_i^{(t-1)}$ in the original $\mathcal{P}^{(t-1)}$ is updated as follows:

$$p_i^{(t)} = \begin{cases} p_i^{(t-1)}, \text{if } i \leq t \\ \pi(p_i^{(t)}|G, \mathcal{C}^{(t)}, \mathcal{P}^{(t-1)}, i), \text{if } i > t \end{cases} \tag{1}$$

where $\pi$ is the adaptation policy of the global plan generator.

By dynamically updating the global plan based on real-time feedback derived from the executor-environment interactions, the agent can promptly assess the validity and efficiency of the current planning strategy and make necessary adjustments accordingly. Furthermore, in cases where the executor deviates from the prescribed plan, the global planner can adaptively revise the course of action to guide the executor toward more effective task execution.

## 2.2 PROGRESSIVE REINFORCEMENT LEARNING

Within the global planning-driven agent paradigms, two key factors influence the overall performance: the quality of the generated global plans, and the degree to which the executor adheres to the plan's directives when interacting with the environment. Accordingly, we employ a three-stage pipeline for training, as shown in the right part of Figure 2.

### 2.2.1 STAGE 1: ENHANCING THE INSTRUCTION ADHERENCE IN EXECUTOR

The ability to comply with the guidance of the global planner is foundational to the entire global planning-guided agent paradigm. Therefore, our focus lies on improving the executor's capacity to follow existing global plans as well as acquire a thorough understanding of the action space $\mathcal{A}$ in the initial training stage. Here we utilize the frontier model, e.g., DeepSeek-V3 (Liu et al., 2024), for the provision of each global plan. Specifically, for each time step $t$:

- **Plan Generation**: We first prompt the model to generate all possible global plans based on the specified task goal $G$ and accumulated contextual information $\mathcal{C}^{(t)}$ to provide a comprehensive set of potential candidates.

- **Plan Selection**: Following this, the model evaluates each of these candidate plans across the dimensions of correctness, executability, format validity, etc., and selects the most suitable one to guide the executor's actions.

In general, the design of the reward function in Stage 1 is *the sum of normalized* components: *format*, *adherence degree*, and ***end-to-end (E2E) performance***.

***Format.*** The model is required to produce its outputs according to a predefined output paradigm. Specifically: *(1)* All responses should be chosen from the two actions, "Thought" or "Action", and must strictly align with the formats of "Thought: ... Action: ..." or "Action: ...". *(2)* The output must be produced in a readable format, without distorted or illegible characters, and then the environmental feedbacks are encapsulated within `<observation>...</observation>` tags. Based on the above requirements, the format reward is defined as:

$$\mathcal{R}_{format} = \begin{cases} 1, \text{if the format is correct} \\ 0, \text{if the format is incorrect} \end{cases} \tag{2}$$

***Adherence Degree.*** This aspect constitutes a core component in fostering the executor's compliance with the global plan during Stage 1. Here we employ a frontier model (e.g., DeepSeek-V3) as the evaluator to score the generated actions. It assesses whether the model's output semantically aligns with the current step of the global plan. Actions are assigned a score of 2 for fully compliant, 1 for partially compliant (e.g., for suggested actions that require the invocation of multiple tools, at least one tool is utilized to support the execution), and 0 for noncompliant actions:

$$\mathcal{R}_{adherence} = \begin{cases} 2, \text{if completely compliant} \\ 1, \text{if partially compliant} \\ 0, \text{if noncompliant} \end{cases} \tag{3}$$

***End-to-End (E2E) Performance.*** The measurement of the first two components concentrates solely on individual execution, rather than assessing the holistic interaction between the agent and the environment. However, in real-world interactions, the problem-solving process may exhibit trajectory redundancy or unintended topic drift, leading to unpredictable deviations from the intended workflow. Therefore, it is essential to obtain a comprehensive, end-to-end view of agent performance in order to assess whether the current interaction trajectory aligns with the expected behavior, and to ensure that the target task is accomplished efficiently and directly, without unnecessary detours.

$$\mathcal{R}_{E2E} = \begin{cases} 2, \text{if accomplished efficiently} \\ 1, \text{if accomplished with redundancy} \\ 0, \text{if unaccomplished} \end{cases} \tag{4}$$

We utilize DeepSeek-V3 to evaluate the end-to-end (E2E) performance $\mathcal{R}_{E2E}$. The agent-environment interactions receive a score of 2 if the task is accomplished in a direct and efficient manner without process redundancy. A score of 1 is assigned if the final task is completed but the interaction involves trajectory redundancy or topic drift. If the agent fails to achieve the final task objective, it is given a score of 0.

### 2.2.2 STAGE 2: CULTIVATING THE CAPACITY OF GLOBAL PLANNER

Following the initial training stage, the agent has acquired a foundational paradigm for global plan following and action execution. In this stage, we shift our focus to enhancing the agent's ability to generate global plans. In generating the global plan, we adopt a *generate-then-select* strategy similar to that used in Stage 1 with the frontier model, which enhances the quality of the global plan ultimately used for explicit guidance, leading to more effective and coherent decision-making. Specifically, all feasible global plans that could potentially solve the given task are first generated, and then the most appropriate one is selected from this pool of candidates. The reward function design in Stage 2 is *the sum of normalized* components: ***format***, ***end-to-end (E2E) performance***, and ***global plan quality***, with the first two already formally defined in Equation (2) and Equation (4).

***Global Plan Quality*** When evaluating the quality of the generated global plan, we consider three primary dimensions: *correctness*, *executability*, and *standardization*. ***(1)*** *Correctness* assesses whether the plan effectively leads to the fulfillment of the task objectives. ***(2)*** *Executability* evaluates the clarity and ease with which the agent can adhere to the instructions, as indicated by the alignment of the executor's action with the global planner's directives. ***(3)*** *Standardization* checks whether the generated instructions conform to a consistent and well-defined format. The quality score of the global plan is calculated as follows:

$$\mathcal{R}_{planning} = R_{correct} + R_{execute} + R_{standard} \tag{5}$$

where $R_{correct}, R_{execute}, R_{standard} \in \{x \in \mathbb{Z} \mid 1 \leq x \leq 5\}$, with 5 indicating the best performance. We use DeepSeek-V3 as the evaluator to score each dimension.

### 2.2.3 Stage 3: Orchestrating the End-to-End (E2E) Performance

Having separately enhanced the model's capabilities in both generating and complying with global plans in earlier stages, Stage 3 focuses on strengthening the coordination between the global planner and the executor, i.e., the joint optimization of our global planning-driven agent paradigm *AdaPlan*. The reward function at this stage is *the sum of normalized **format** and **end-to-end (E2E) performance***, which directly prioritizes comprehensive performance of the ultimate task objective.

## 3 Experiments

### 3.1 Experimental Setup

**Datasets.** During *training*, we collect data from the training splits of four datasets: ALFWorld (Shridhar et al., 2021), IQA (Gordon et al., 2018), TextCraft (Prasad et al., 2024), and Wordle (Abdulhai et al., 2023). Our *evaluation* is conducted on six benchmarks. We employ the test splits of ALFWorld, IQA, TextCraft, and Wordle for in-domain (ID) assessment, and the full dataset samples of MAZE (Abdulhai et al., 2023) and BabyAI (Chevalier-Boisvert et al., 2019) for out-of-domain (OOD) scenarios. We collected data from prior work (Song et al., 2024; Xi et al., 2024), and use only the task instructions and their corresponding final answers for RL-related training and evaluation, with the overall statistics and details of the datasets described in Table 5 and Section B.1. In this work, we adopt the *LLM-as-Judge* (Zheng et al., 2023; Gu et al., 2024) paradigm to verify the model's end-to-end (E2E) performance, including ***(1)*** the task completion rates, and ***(2)*** the efficiency of interaction trajectories, and then calculate the average scores as the evaluation metric.

**Models and Implementation.** We validate the effectiveness of PilotRL across different base and instruction-tuned models, including Qwen2.5-7B-Instruct (Yang et al., 2024), LLaMA3.1-8B-Instruct (Dubey et al., 2024), and Qwen3-8B (Yang et al., 2025). The *reinforcement learning (RL)* framework is built on verl (Sheng et al., 2025) with Group Relative Policy Optimization (GRPO) (Shao et al., 2024) as the learning algorithm. The total training dataset contains 5725 samples. Each sample undergoes 16 rollouts, with a training batch size of 256 and a rollout batch size of 64. The total number of training epochs is set to 4, with 1 epoch allocated to Stage 1, 2 epochs to Stage 2, and an additional 1 epoch dedicated to Stage 3. The learning rate is set at 1e-6. Following the approach proposed by Sun et al. (2025), we employ the frontier model DeepSeek-V3 to simulate real-world environmental behaviors. Notably, in our training setup, the environmental observation $\mathcal{O}$ is concatenated into the interaction process, which are not generated by the training policy. To prevent these segments from influencing gradient updates, we apply masking during loss calculation, where we mask out all content enclosed within `<observation>...</observation>` tags. When conducting *supervised fine-tuning (SFT)* as baseline competitors, we utilized a learning rate scheduler featuring linear warm-up and cosine decay, peaking at a learning rate of 2e-5, alongside a warmup ratio of 0.03 and a weight decay of 0.0 and a batch size of 256 for 4 epochs.

**Baselines.** We compare PilotRL with the following baselines: (1) We employ GPT-4o and GPT-4o-mini (Hurst et al., 2024) as the ***Close-Sourced Models*** competitors. (2) ***Open-Sourced Agent-Specific Models*** include Agent-FLAN-7B (Chen et al., 2024), LLaMA-xLAM-2-8B-fc-r (Zhang et al., 2024a) and DeepResearcher-7B (Zheng et al., 2025). (3) The simplest baseline is ***Naive Response***, where the model generates responses directly without any training or prompting strategies. (4) ***ReAct*** (Yao et al., 2023) is the common agent paradigm that prompts agents to integrate single-step reasoning with immediate action execution. (5) ***MPO*** (Xiong et al., 2025) acts as an external

Table 1: Comparison of PilotRL with baselines. "w/o Plan." indicates whether the inference paradigm includes global planning as a mechanism for providing explicit guidance. The best and second best of each model are in **bold** and underlined.

| Backbone Model | Method | w/o Plan. | ALFWorld | IQA | TextCraft | Wordle | BabyAI | MAZE | Avg. |
|---|---|---|---|---|---|---|---|---|---|
| | | | | | *In-Domain (ID)* | | *Out-of-Domain (OOD)* | | |
| ***Close-Sourced Models*** | | | | | | | | | |
| GPT-4o | – | ✗ | 75.83 | 66.59 | 68.50 | 78.65 | 57.87 | 60.42 | 67.98 |
| GPT-4o-mini | – | ✗ | 52.35 | 40.32 | 46.74 | 42.51 | 43.96 | 34.36 | 45.21 |
| ***Open-Sourced Agent-Specific Models*** | | | | | | | | | |
| Agent-FLAN-7B | – | ✗ | 70.54 | 57.62 | 24.66 | 22.28 | 24.39 | 28.93 | 38.07 |
| LLaMA-xLAM-2-8B-fc-r | – | ✗ | 50.38 | 53.74 | 46.15 | 48.52 | 54.26 | 36.57 | 48.27 |
| DeepResearcher-7B | – | ✗ | 58.36 | 62.87 | 55.58 | 47.17 | 52.75 | 40.82 | 52.93 |
| ***Open-Sourced Base / Instruct Models*** | | | | | | | | | |
| Qwen2.5-7B-Instruct | Naive Response | ✗ | 48.78 | 35.40 | 30.35 | 34.72 | 40.39 | 33.80 | 37.24 |
| | ReAct | ✗ | 52.15 | 37.57 | 34.46 | 40.43 | 44.08 | 37.52 | 41.04 |
| | + MPO | ✔ | 67.31 | 58.64 | 52.28 | 56.76 | 53.85 | 49.67 | 56.42 |
| | SFT | ✔ | 67.53 | 63.35 | 73.10 | 74.64 | 55.68 | 46.92 | 63.54 |
| | Vanilla RL | ✗ | 65.49 | 64.78 | 70.76 | 71.28 | 58.62 | 50.59 | 63.59 |
| | **PilotRL (ours)** | ✔ | **70.80** | **67.84** | **75.37** | **77.69** | **61.56** | **57.93** | **68.53** |
| LLaMA3.1-8B-Instruct | Naive Response | ✗ | 35.63 | 38.56 | 38.22 | 36.40 | 46.17 | 30.64 | 37.60 |
| | ReAct | ✗ | 38.48 | 42.94 | 45.83 | 38.56 | 47.36 | 36.92 | 41.68 |
| | + MPO | ✔ | 54.25 | 50.31 | 43.86 | 52.60 | 58.92 | 45.33 | 50.88 |
| | SFT | ✔ | 74.92 | 69.84 | 58.42 | 73.55 | 55.52 | 50.76 | 63.84 |
| | Vanilla RL | ✗ | 70.68 | 68.13 | 60.57 | 68.80 | 59.74 | 52.05 | 63.33 |
| | **PilotRL (ours)** | ✔ | **78.53** | **72.78** | **64.76** | **79.61** | **68.24** | **58.68** | **70.43** |
| Qwen3-8B | Naive Response | ✗ | 54.08 | 42.14 | 36.37 | 34.95 | 48.46 | 36.53 | 42.09 |
| | ReAct | ✗ | 62.56 | 50.58 | 44.62 | 41.60 | 54.35 | 42.68 | 49.40 |
| | + MPO | ✔ | 65.42 | 54.67 | 46.25 | 48.79 | 56.81 | 39.50 | 51.91 |
| | SFT | ✔ | 64.73 | 62.75 | 63.16 | 75.83 | 59.67 | 49.25 | 62.57 |
| | Vanilla RL | ✗ | 68.47 | **70.29** | 67.35 | 80.42 | 63.44 | 52.04 | 67.00 |
| | **PilotRL (ours)** | ✔ | **72.51** | 69.06 | **71.48** | **83.65** | **65.28** | **56.62** | **69.77** |

plug-and-play planner that endows the model with meta-plans to provide explicit guidance during task execution. (6) We also perform ***Supervised Fine-Tuning (SFT)*** on models, a widely adopted training strategy in a series of previous works (Chen et al., 2024; Song et al., 2024; Xi et al., 2024; Zeng et al., 2024; Zhang et al., 2024b; Fu et al., 2025). Specifically, we utilize frontier models (e.g., DeepSeek-V3) to generate global plans that guide the execution of target tasks. (7) ***Vanilla RL*** is the naive reinforcement learning process that utilizes the Group Relative Policy Optimization (GRPO) (Shao et al., 2024) algorithm. In this setup, we utilize only the format and end-to-end (E2E) performance as the reward metrics. Details are discussed in Section B.2.

## 3.2 MAIN RESULTS

The main results of baselines and PilotRL are demonstrated in Table 1, and we show the methods based on models with base and instruction-tuned versions respectively. From the main results, we summarize the observations below.

**PilotRL is effective across different models.** Experimental results in Table 1 show that our PilotRL consistently outperforms other baseline approaches on both base and instruction-tuned models in terms of agent task completion. Compared to the *naive response*, PilotRL enhances the average downstream task performances by 78.51%. Remarkably, when compared to *open-sourced agent-specific models* such as DeepResearcher-7B, our approach achieves over 29.47% higher performance with the same backbone model of Qwen2.5-7B-Instruct. In comparison to the plug-and-play external planner *MPO*, our method achieves an average improvement of 31.10%, further highlighting the importance of tight coordination between the planner and executor in effectively solving agent-oriented tasks. Furthermore, open-sourced models enhanced with PilotRL demonstrate the potential to outperform *close-sourced proprietary models* in agent problem-solving. Specifically, models integrated with PilotRL achieve an average improvement of 2.35% over GPT-4o, while showing a more substantial gain of 53.90% over GPT-4o-mini at a comparable parameter scale.

**AdaPlan paradigm + RL boosts agent performances.** Here we focus on analyzing the performance of two baseline methods: *SFT* and *Vanilla RL*. The primary distinction between SFT and PilotRL lies in the training strategies, while the key difference between Vanilla RL and our method

is whether to incorporate the AdaPlan paradigm to provide global guidance for agent execution. As presented in Table 1, the average performance of SFT and Vanilla RL is quite similar on both Qwen2.5-7B-Instruct and LLaMA3.1-8B-Instruct. This suggests that the enhancement brought by global plan guidance in SFT is roughly on par with the incremental gain achieved through RL-based training. Specifically, for in-domain (ID) tasks, SFT outperforms Vanilla RL by a marginal average of 2.75%, whereas Vanilla RL achieves an average lead of 5.80% in out-of-domain (OOD) tasks. For reasoning-oriented models such as Qwen3-8B, which inherently possess a certain degree of multi-step reasoning and decision-making capabilities required for complex agent tasks, the performance gains from the AdaPlan paradigm are insufficient to offset the advantages of RL over SFT training. In contrast, PilotRL demonstrates robust performance gains across models with diverse characteristics, achieving consistent improvements over both SFT and Vanilla RL by 9.89% and 7.64%, respectively. These observations further highlight the importance of combining the global planning capabilities of the AdaPlan paradigm with RL training, as embodied in our PilotRL framework, for enhancing model performance in complex agent scenarios.

# 4 ABLATIONS AND ANALYSIS

We conduct ablation studies on open-sourced models to highlight the contribution of each training stage and to assess the impact of their sequential order on PilotRL. Furthermore, we perform an in-depth analysis of PilotRL's effectiveness, examining key aspects such as our AdaPlan paradigm for explicit guidance, the architecture of unified planner-executor, and the co-evolution of components.

## 4.1 TRAINING STAGE ABLATION

**Necessity of Progressive Training.** We aggregated the reward functions from all training stages to verify the importance of incrementally optimizing the planning and execution capabilities in a staged and progressive manner. Results are presented in Table 2 (1 & 2 & 3), where we observe a performance drop of 3.32% compared to our multi-stage training strategy (1 → 2 → 3). A primary cause of this performance drop lies in the intrinsic complexity and potential conflicts among heterogeneous reward signals. Specifically, the planning-oriented and execution-driven components exert distinct behavioral demands on the model, which can lead to unstable policy updates during training. For instance, in the early stages of training, the model may lack a sufficiently mature structure for guidance follow-up, making it difficult to accurately adhere to global plans. It results in conflicting gradient signals and ultimately reduces learning efficiency.

Table 2: Analysis of the training stages and sequential order. "Order" refers to the sequence of Stage 1, 2, and 3. "1 & 2 & 3" denotes a training setting in which the reward functions from all stages are applied simultaneously. We compute the *average* performance of the evaluated models across each benchmark. The best and second best scores are in **bold** and underlined.

| Order | In-Domain | Out-of-Domain | Avg. |
|---|---|---|---|
| *Standard Pipeline* | | | |
| **1 → 2 → 3** | **73.68** | **61.39** | **69.58** |
| *Necessity of Progressive Training* | | | |
| 1 & 2 & 3 | 71.64 (↓ 2.77%) | 58.52 (↓ 4.68%) | 67.27 (↓ 3.32%) |
| *The Role of Each Stage* | | | |
| 2 → 3 | 70.82 (↓ 3.88%) | 58.33 (↓ 4.98%) | 66.66 (↓ 4.20%) |
| 1 → 3 | 70.66 (↓ 4.10%) | 58.39 (↓ 4.89%) | 66.57 (↓ 4.33%) |
| 1 → 2 | 72.21 (↓ 2.00%) | 59.02 (↓ 3.86%) | 67.81 (↓ 2.54%) |
| *Sequential Order of Stages* | | | |
| 2 → 1 → 3 | 72.79 (↓ 1.21%) | 59.88 (↓ 2.46%) | 68.48 (↓ 1.58%) |

**The Role of Each Stage.** To assess the contribution of each individual stage, we conduct three ablation studies by sequentially removing Stage 1, 2, and 3, respectively. The models are then evaluated on both in-domain (ID) and out-of-domain (OOD) benchmark tasks, with the results presented in Table 2. To ensure a fair comparison and control for the impact of training data volume on performance, we fix the total number of training epochs at 4, which is consistent with the main experimental setup, and allocate 2 epochs to each of the remaining two stages for training.

- *Removing Stage 1.* Stage 1 is designed to strengthen the models' ability to follow instructions when performing agent tasks. As shown in Table 2 (2 → 3), the removal of Stage 1 results in

a performance drop of 4.20% in overall model performance. This decline occurs because Stage 1 acts as the cornerstone for Stage 2. Without robust instruction-following behavior, the model struggles to adhere to the provided global plans, which are essential for delivering explicit guidance. As a result, the effectiveness of subsequent training stages is diminished to a certain extent.

- *Removing Stage 2.* Building on Stage 1, Stage 2 focuses on optimizing the quality of generated global plans, thereby providing more effective high-level guidance for complex agent tasks. As indicated in Table 2 (1 → 3), eliminating Stage 2 results in a modest decline of 4.33% in performance relative to the model trained with all three stages.

- *Removing Stage 3.* Stage 3 aims to optimize the coordination between the global planner and executor, thereby enhancing the model's end-to-end performance in agent tasks. As observed in Table 2 (1 → 2), excluding Stage 3 leads to a performance drop of 2.54%. Nevertheless, owing to the presence of fully implemented Stage 1 and Stage 2, the performance gap relative to the model trained through all three stages remains narrow and relatively small.

**Sequential Order of Stages.** We swap Stage 1 and Stage 2 to evaluate their influence on model performance. As seen in Table 2 (2 → 1 → 3), such reordering results in a slight performance decline of 1.58%. It supports the robustness of our original training sequence, which prioritizes the development of guidance-following capabilities before refining global plan generation skills. It is grounded in the need for a strong foundation of instruction follow-up to enhance the quality of global plans. Only with this foundation can the model make meaningful strides in developing its ability to generate global plans that effectively guide the action execution during agent task completion.

## 4.2 FURTHER ANALYSIS

**AdaPlan vs. ReAct.** We compare the performance of the AdaPlan and ReAct paradigms in agent tasks. Neither of these paradigms undergoes additional training, with distinct prompt strategies employed to induce different thinking patterns in the model instead. As presented in Table 3, the experimental results indicate that our proposed AdaPlan exhibits greater efficacy in enabling the model to accomplish complex agent tasks by leveraging global planning as guidance, which outperforms ReAct by 12.76%.

Table 3: Analysis on the agent paradigms of *AdaPlan* and *ReAct* on In-Domain (ID) and Out-of-Domain (OOD) tasks. The best score of each model are in **bold**.

| Backbone Model | Paradigm | ID | OOD | Avg. |
|---|---|---|---|---|
| Qwen2.5-7B -Instruct | AdaPlan | **50.54** | **44.98** | **48.69** |
| | ReAct | 41.15 | 40.80 | 41.04 |
| LLaMA3.1-8B -Instruct | AdaPlan | **47.42** | **47.20** | **47.34** |
| | ReAct | 41.45 | 42.14 | 41.68 |
| Qwen3-8B | AdaPlan | **53.69** | **51.49** | **52.95** |
| | ReAct | 49.84 | 48.52 | 49.40 |

**Unified Architecture vs. Isolated Planner-Executor Architecture.** We conduct an evaluation against the isolated planner and executor framework (Erdogan et al., 2025) to validate the effectiveness of integrating both components within a unified model architecture. In the isolated architecture setting, we employ the same backbone model and separately train the planner and executor modules following the Stage 1 and Stage 2 RL procedures described in PilotRL,

Table 4: Analysis of the *unified* and *isolated* planner-executor architectures on In-Domain (ID) and Out-of-Domain (OOD) tasks. The best scores are in **bold**.

| Backbone Model | Architecture | ID | OOD | Avg. |
|---|---|---|---|---|
| Qwen2.5-7B -Instruct | Unified | **72.93** | **59.75** | **68.53** |
| | Isolated | 68.94 | 55.18 | 64.36 |
| LLaMA3.1-8B -Instruct | Unified | **73.92** | **63.46** | **70.43** |
| | Isolated | 68.68 | 59.18 | 65.51 |
| Qwen3-8B | Unified | **74.18** | **60.95** | **69.77** |
| | Isolated | 72.66 | 56.02 | 67.11 |

with each component trained for 2 epochs. As summarized in Table 4, the isolated architecture suffers from a performance drop of 5.63% compared to the unified architecture, in which both functionalities are learned jointly in an end-to-end manner, further emphasizing the importance of co-developing planning and execution capabilities within a single model.

**How planner, executor, and their coordination co-evolve during agent learning?** We analyze the evolution of reward scores for the global planner, the executor, and the end-to-end (E2E) performance in the training process of LLaMA3.1-8B-Instruct. As shown in Figure 3, the executor's ability of plan adhesion saw a marked improvement during Stage 1 and remained stable with slight growth in subsequent stages. The global planner's performance, which generates high-level plans for explicit guidance, exhibits a notable improvement in Stage 2 (epoch 2 & 3). It experiences a mild

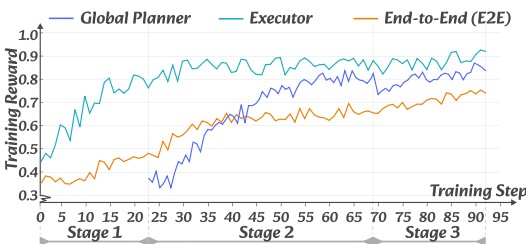

Figure 3: Normalized rewards for global planner, executor and end-to-end (E2E) performance in training LLaMA3.1-8B-Instruct.

decline at the beginning of Stage 3, followed by a continuous upward trend. We speculate that this temporary drop reflects an adaptation period, during which the planner adjusts its generation to better align with the executor's capabilities. Meanwhile, the E2E reward increases steadily throughout the entire training process, indicating a consistent improvement in the system's overall performance.

## 5 RELATED WORK

**LLM as Agent** The emergence of Large Language Models (LLMs) has driven research into the development of LLM-based agent systems (Zeng et al., 2024). The most common paradigm for LLM-based agent systems is ReAct (Yao et al., 2023), which integrates Chain-of-Thought (CoT) reasoning with agent actions in an interleaved manner to accomplish multiple tasks. However, this step-by-step reasoning framework struggles in scenarios demanding long-term strategic planning or complex multi-step coordination, e.g., household exploration (Shridhar et al., 2021) and games involving foresighted planning (Abdulhai et al., 2023), which highlights a pressing need for mechanisms with coherent and long-term planning. Even though there have been efforts aimed to incorporate explicit guidance into agent task completion (Deng et al., 2023; Zeng et al., 2024), the planner and executor are typically implemented in isolated architectural frameworks, leading to suboptimal instruction generation and execution alignment. Moreover, although closed-source proprietary models often demonstrate strong performance in agent tasks, open-source models generally fall short in comparison (Liu et al., 2023). While recent studies have tried to collect expert trajectories from frontier LLMs (e.g., GPT-4) to fine-tune open-sourced models (Chen et al., 2023; 2024; Song et al., 2024; Zeng et al., 2024; Zhang et al., 2024b), such behavioral cloning strategy hinders the model's generalization performance on out-of-distribution tasks. Therefore, it is necessary to introduce a more flexible training framework to cultivate models' intrinsic generalization capabilities, e.g., reinforcement learning (RL).

**Reinforcement Learning in LLMs** Compared to the supervised fine-tuning (SFT), reinforcement learning (RL) provides a more powerful paradigm for training LLM-based agents which are capable of autonomous planning, decision-making, and environmental interaction without explicit supervision (Guo et al., 2025; Jaech et al., 2024; Team et al., 2025). Among all the RL algorithms, GRPO (Shao et al., 2024; Guo et al., 2025) is specifically designed for LLMs, which has proven to be highly effective by replacing the traditional critic with a group-based evaluation strategy. Efforts have been made to enhance the agent capability in LLMs through the RL process, with notable works for information retrieval tasks (Jin et al., 2025; Song et al., 2025) and tool utilization scenarios (Feng et al., 2025; Li et al., 2025). We situate our research on agent capability enhancement within the RL landscape for its effectiveness in fostering exploration and the emergence of novel strategies, and shift away from the commonly used ReAct framework (Yao et al., 2023), toward a global-plan-driven paradigm that supports more strategic and forward-looking decision-making.

## 6 CONCLUSION

In this paper, we introduce AdaPlan, an adaptive global plan-based agent paradigm. Based on the proposed paradigm, we put forward PilotRL, a global planning-guided training framework for LLM agents driven by progressive reinforcement learning. Experimental results indicate that PilotRL achieves excellent training outcomes in agent scenarios.

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

APPENDIX

## A  GROUP RELATIVE POLICY OPTIMIZATION (GRPO)

We utilize the Group Relative Policy Optimization (GRPO) as the RL algorithm. For each question $x \sim \mathcal{D}$, the behavior policy $\pi_{\theta_{\text{old}}}$ generates a set of $G$ candidate completions $\tau = \{y_i\}_{i=1}^{G} \sim \pi_{\theta_{\text{old}}}(\cdot|x)$, with each response receiving a scalar reward $r_i$. The training objective is to optimize the policy $\pi_\theta$ based on reference policy $\pi_{\theta_{\text{ref}}}$:

$$\mathcal{J}(\theta) = \mathbb{E}_{x \sim \mathcal{D}, \{y_i\}_{i=1}^{G} \sim \pi_{\theta_{\text{old}}}(\cdot|x)} \frac{1}{G} \sum_{i=1}^{G} [\min(\frac{\pi_\theta(y_i|x)}{\pi_{\theta_{\text{old}}}(y_i|x)}\hat{A}_i,$$
$$\text{clip}(\frac{\pi_\theta(y_i|x)}{\pi_{\theta_{\text{old}}}(y_i|x)}, 1-\epsilon, 1+\epsilon)\hat{A}_i) - \beta\mathbb{D}_{\text{KL}}(\pi_\theta||\pi_{\theta_{\text{ref}}})] \quad (6)$$

where the group-normalized advantage $\hat{A}_i$ of the $i$-th rollout in current group is defined as:

$$\hat{A}_i = \frac{r_i - \text{mean}(\{r_j\}_{j=1}^{G})}{\text{std}(\{r_j\}_{j=1}^{G})}$$

An overview of the GRPO algorithm is illustrated in Figure 4. In this formulation, $\epsilon$ denotes the clipping ratio, a hyperparameter that controls the allowable deviation between the updated and reference policies. The `clip` function restricts the importance weight $r_i$ within the range $[1-\epsilon, 1+\epsilon]$, which enhances training stability and reduces the risk of policy collapse. The parameter $\beta$ represents the Kullback–Leibler (KL) loss coefficient (Hall, 1987), which governs the strength of the KL divergence penalty included in the objective function. This penalty term helps constrain the policy updates, ensuring that the learned policy remains sufficiently close to the original reference policy and thereby improving overall training stability.

## B  EXPERIMENT DETAILS

### B.1  DATASETS

To evaluate the performance of PilotRL , we conduct experiments using six datasets for agent tasks. Specifically, four datasets are used for training and in-domain (ID) performance evaluation, while the remaining two are reserved for out-of-domain (OOD) assessment, as shown in Table 5.

- **ALFWorld** (Shridhar et al., 2021): ALFWorld is a home-oriented environment built upon TextWorld, where agents are required to navigate through rooms and apply common sense

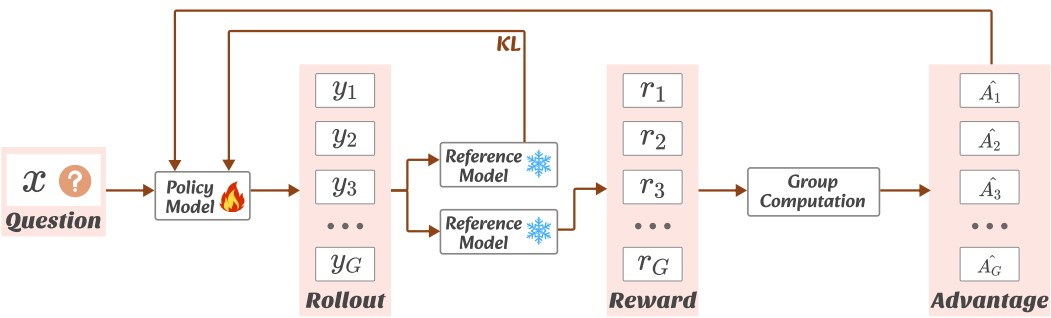

Figure 4: An illustration for the Group Relative Policy Optimization (GRPO) pipeline.

Table 5: Statistics of datasets for training and evaluation.

| Classification | Dataset | #Training Num. | #Testing Num. |
|---|---|---|---|
| *In-Domain* | ALFWorld | 3000 | 321 |
| | IQA | 1465 | 162 |
| | TextCraft | 400 | 74 |
| | Wordle | 860 | 95 |
| *Out-of-Domain* | BabyAI | – | 400 |
| | MAZE | – | 215 |

reasoning to perform various tasks. It mirrors the embodied settings found in the ALFRED dataset (Shridhar et al., 2020), and offers human-annotated ideal trajectories for use in imitation learning.

- **IQA** (Gordon et al., 2018): The Interactive QA dataset is a question answering task in which an agent need to engage with a dynamic visual environment to find answers. Here we utilize the text version from Jia et al. (2024).

- **TextCraft** (Prasad et al., 2024): It is a text-only environment for crafting Minecraft items that resembles cooking recipes with steps of varying complexity. This dataset exhibits an inherently decomposable structure, providing a more suitable environment for our proposed paradigm.

- **Wordle** (Abdulhai et al., 2023): It is a word-guessing game designed to assess agents' reasoning capabilities at the letter level, where the agents attempt to identify a target word selected from a predefined vocabulary consisting of five-letter words. In order to successfully identify the target word with minimum trials within the limited number of allowed attempts, it is crucial for the model to employ efficient global planning.

- **MAZE** (Abdulhai et al., 2023): MAZE is also a word-based puzzle game in which agents, serving as players, are aware of their current position, the location of the goal, and the presence of walls in the four cardinal directions (up, down, left, and right).

- **BabyAI** (Chevalier-Boisvert et al., 2019): The BabyAI dataset evaluates agent performance in embodied navigation and interaction scenarios. It features a simulated grid-world environment containing 40 instruction-following tasks, where agents are required to understand commands and interact with objects accordingly.

We collected the training and evaluation data from Song et al. (2024) and Xi et al. (2024). For ALFWorld and IQA, we utilize the datasets as provided in Song et al. (2024), while for TextCraft, Wordle, MAZE, and BabyAI, we adopt the versions from Xi et al. (2024). The reference trajectories included in these original data sources are used exclusively for supervised fine-tuning (SFT) of the baselines. During both the reinforcement learning (RL) training and evaluation phases, we only make use of the task instructions and their corresponding final answers.

## B.2 BASELINES

In this section, we provide a comprehensive overview of the various methods that serve as baselines in our comparison.

- **Close-Sourced Models**: Closed-source models are considered to represent the current state-of-the-art in LLM capabilities and are regarded as the most competitive baseline methods. We have selected GPT-4o and GPT-4o-mini (Hurst et al., 2024) to assess the upper bound of the model performance on agent tasks.

- **Open-Sourced Agent-Specific Models**: These models refer to models that were trained specifically on agent-task datasets. We have selected Agent-FLAN-7B (Chen et al., 2024), LLaMA-xLAM-2-8B-fc-r (Zhang et al., 2024a) and DeepResearcher-7B (Zheng et al., 2025) to represent the open-sourced agent-specific models for comparison to assess PilotRL's relative advantages. Specifically, the backbone model of DeepResearcher-7B is Qwen2.5-7B-Instruct (Yang et al., 2024), which facilitates a more direct comparison with Qwen2.5-7B-Instruct + PilotRL.

- **Naive Response**: It refers to the case where the model directly generates responses without any training (e.g., SFT, RL, etc.) or prompting (e.g., ReAct) strategies.

- **ReAct** (Yao et al., 2023): It is the prompting strategy that integrates single-step reasoning with the execution of the current action, which is a common agent paradigm.

- **MPO** (Xiong et al., 2025): The Meta Plan Optimization (MPO) framework improves the agent's planning capabilities by integrating explicit guidance into the decision-making process. As an external plug-and-play planner, MPO provides the model with high-level meta-plans that serve as structured guidance during task execution. One key distinction between MPO and PilotRL lies in the integration and training of the planner and executor components. In our approach, both the planner and executor reside within the same model and are trained jointly. In contrast, MPO maintains separate models for planning and execution, where only the planner is trained while the executor's parameters remain frozen, leading to limited coordination between the two components.

- **Supervised Fine-Tuning** (SFT): This training strategy is widely adopted in a series of studies (Chen et al., 2024; Song et al., 2024; Xi et al., 2024; Zeng et al., 2024; Zhang et al., 2024b; Fu et al., 2025). However, existing studies have shown that compared to RL, SFT generally exhibits weaker generalization capabilities on new tasks—particularly when the training data consists of multi-step trajectories for problem-solving (Shao et al., 2024; Team et al., 2025). This is because such trajectories may contain redundant or suboptimal paths to task completion. Moreover, SFT tends to bias the model toward previously seen execution paths, limiting its ability to adapt or generalize to novel scenarios through compositional or analogical reasoning. During SFT, we use the same datasets with PilotRL. In addition, we incorporate the original agent-environment interaction trajectories into training, a setting that differs from **Vanilla RL** and our **PilotRL**. Furthermore, we generate global plans for guiding task completion using DeepSeek-V3, and feed both the interaction trajectories and the corresponding global plans into the model during training. This setup allows us to compare PilotRL over existing baselines under a more fair and controlled experimental condition.

- **Vanilla RL**: We also conduct training with the naive reinforcement learning process utilizing the Group Relative Policy Optimization (GRPO) (Shao et al., 2024) algorithm. Here we employ only the format and end-to-end (E2E) performance as the reward metrics. This baseline is for validating the effectiveness of adaptive global planning.

## B.3 ABLATION STUDY AND FURTHER ANALYSIS DETAILS

In this section, we report the original performance scores of the models on each benchmark during training stage and training sequence ablation, as well as reward design ablation, as depicted in Table 6, Table 7 and Table 8.

**Declaration for *Figure 3*** It is worth noting that when analyzing the evolution of reward scores for the global planner, the executor, and the end-to-end (E2E) performance using LLaMA3.1-8B-

Table 6: **Original scores for each benchmark of the ablation study on multiple training stages and sequential order.** It is the detailed version of Table 2 . "Order" is the sequential order of Stage 1, 2, and 3 during training. Specifically, "1 & 2 & 3" refers to a joint training configuration in which reward functions from all three stages are merged and optimized concurrently, where the target model generates global plans independently throughout the entire training process. The best and second best scores of each model are in **bold** and underlined.

| Order | Backbone Model | ALFWorld | IQA | TextCraft | Wordle | BabyAI | MAZE | Avg. |
|---|---|---|---|---|---|---|---|---|
| | | In-Domain (ID) | | | | Out-of-Domain (OOD) | | |
| *Standard Pipeline* | | | | | | | | |
| **1 → 2 → 3** **(ours)** | Qwen2.5-7B-Instruct | **70.80** | **67.84** | 75.37 | 77.69 | **61.56** | **57.93** | **68.53** |
| | LLaMA3.1-8B-Instruct | **78.53** | 72.78 | **64.76** | **79.61** | **68.24** | **58.68** | **70.43** |
| | Qwen3-8B | 72.51 | 69.06 | **71.48** | **83.65** | **65.28** | **56.62** | **69.77** |
| *Necessity of Progressive Training* | | | | | | | | |
| 1 & 2 & 3 | Qwen2.5-7B-Instruct | 68.29 | 65.43 | 72.91 | 75.82 | 57.98 | 54.37 | 65.80 |
| | LLaMA3.1-8B-Instruct | 75.56 | 70.42 | 63.03 | 74.51 | 63.74 | 56.00 | 67.21 |
| | Qwen3-8B | 70.89 | 71.30 | 69.68 | 81.84 | 63.19 | 55.81 | 68.79 |
| *Effectiveness of Stage 1 (Instruction Adherence)* | | | | | | | | |
| 2 → 3 | Qwen2.5-7B-Instruct | 66.37 | 63.85 | 72.16 | 74.93 | 60.05 | 52.54 | 64.98 |
| | LLaMA3.1-8B-Instruct | 73.86 | 70.19 | 63.75 | 72.66 | 64.37 | 54.93 | 66.63 |
| | Qwen3-8B | 70.97 | 69.63 | 70.12 | 81.35 | 63.96 | 54.10 | 68.36 |
| *Effectiveness of Stage 2 (Global Planner Cultivation)* | | | | | | | | |
| 1 → 3 | Qwen2.5-7B-Instruct | 66.72 | 66.38 | 71.74 | 76.56 | 58.85 | 53.48 | 65.62 |
| | LLaMA3.1-8B-Instruct | 73.04 | 72.43 | 61.59 | 70.47 | 66.32 | 53.26 | 66.19 |
| | Qwen3-8B | 70.56 | 68.36 | 69.04 | 80.98 | 64.47 | 53.95 | 67.89 |
| *Effectiveness of Stage 3 (Dual-Process Collaboration)* | | | | | | | | |
| 1 → 2 | Qwen2.5-7B-Instruct | 67.49 | 65.82 | **75.65** | 73.34 | 60.78 | 53.17 | 66.04 |
| | LLaMA3.1-8B-Instruct | 75.40 | 71.55 | 62.88 | 75.67 | 65.19 | 56.92 | 67.94 |
| | Qwen3-8B | 72.18 | 72.61 | 70.59 | 83.27 | 64.73 | 53.28 | 69.44 |
| *Sequential Order of Stages* | | | | | | | | |
| 2 → 1 → 3 | Qwen2.5-7B-Instruct | 70.12 | 66.08 | 73.98 | **77.85** | 59.63 | 55.67 | 67.22 |
| | LLaMA3.1-8B-Instruct | 77.25 | **73.15** | 64.02 | 77.63 | 65.98 | 58.14 | 69.36 |
| | Qwen3-8B | **72.94** | **73.86** | 68.55 | 78.02 | 65.07 | 54.80 | 68.87 |

Table 7: **Original scores for each benchmark of the agent paradigm analysis.** It is the detailed version of Table 3. The best scores of each model are in **bold**. It shows that *AdaPlan* consistently outperforms *ReAct* on both in-domain and out-of-domain agent tasks across all models, demonstrating performance gains of 18.64%, 13.58%, 7.19% on Qwen2.5-7B-Instruct, LLaMA3.1-8B-Instruct, and Qwen3-8B, respectively.

| Backbone Model | Paradigm | ALFWorld | IQA | TextCraft | Wordle | BabyAI | MAZE |
|---|---|---|---|---|---|---|---|
| | | In-Domain (ID) | | | | Out-of-Domain (OOD) | |
| Qwen2.5-7B -Instruct | ReAct | 52.15 | 37.57 | 34.46 | 40.43 | 44.08 | 37.52 |
| | AdaPlan | **59.72** (↑ 14.52%) | **43.68** (↑ 16.26%) | **45.54** (↑ 32.15%) | **53.23** (↑ 31.66%) | **47.90** (↑ 8.67%) | **42.05** (↑ 12.07%) |
| LLaMA3.1-8B -Instruct | ReAct | 38.48 | 42.94 | 45.83 | 38.56 | 47.36 | 36.92 |
| | AdaPlan | **44.19** (↑ 14.84%) | **48.02** (↑ 11.83%) | **46.67** (↑ 1.83%) | **50.78** (↑ 31.69%) | **54.46** (↑ 14.99%) | **39.94** (↑ 8.18%) |
| Qwen3-8B | ReAct | 62.56 | 50.58 | 44.62 | 41.60 | 54.35 | 42.68 |
| | AdaPlan | **63.34** (↑ 1.25%) | **53.82** (↑ 6.41%) | **44.98** (↑ 0.81%) | **52.61** (↑ 26.47%) | **55.73** (↑ 2.54%) | **47.24** (↑ 10.68%) |

Instruct + PilotRL, we normalized all reward scores to the range $[0, 1]$ for visualization and comparison purposes. The reward metrics include:

- **Global Planner**: This reward function (Equation (5)) is introduced starting from Stage 2, and operates during Stage 2 (epoch 2 & 3). In Stage 3, we only evaluate and record this metric without using it for model updates.

- **Executor**: This reward (Equation (3)) is used as the training objective solely in Stage 1. In subsequent stages, we continue to log its value for analysis, but it no longer influences model updates.

Table 8: **Original scores for each benchmark of the planner-executor architecture analysis.** It is the detailed version of Table 4. The best scores of each model are in **bold**. It shows that the *unified architecture* consistently outperforms *isolated architecture* on both in-domain and out-of-domain agent tasks across all models, with measured improvements of 6.48%, 7.51%, 3.96% on Qwen2.5-7B-Instruct, LLaMA3.1-8B-Instruct, and Qwen3-8B, respectively.

| Backbone Model | Architecture | ALFWorld | IQA | TextCraft | Wordle | BabyAI | MAZE |
|---|---|---|---|---|---|---|---|
| | | | *In-Domain (ID)* | | | *Out-of-Domain (OOD)* | |
| Qwen2.5-7B -Instruct | Isolated | 68.85 | 64.18 | 72.60 | 70.14 | 58.29 | 52.07 |
| | Unified | **70.80** (↑ 2.83%) | **67.84** (↑ 5.70%) | **75.37** (↑ 3.82%) | **77.69** (↑ 10.76%) | **61.56** (↑ 5.61%) | **57.93** (↑ 11.25%) |
| LLaMA3.1-8B -Instruct | Isolated | 71.87 | 70.83 | 60.96 | 71.05 | 62.71 | 55.64 |
| | Unified | **78.53** (↑ 9.27%) | **72.78** (↑ 2.75%) | **64.76** (↑ 6.23%) | **79.61** (↑ 12.05%) | **68.24** (↑ 8.82%) | **58.68** (↑ 5.46%) |
| Qwen3-8B | Isolated | 71.74 | 67.71 | 68.96 | 82.23 | 60.55 | 51.49 |
| | Unified | **72.51** (↑ 1.07%) | **69.06** (↑ 1.99%) | **71.48** (↑ 3.65%) | **83.65** (↑ 1.73%) | **65.28** (↑ 7.81%) | **56.62** (↑ 9.96%) |

- **End-to-End (E2E) Performance**: The reward based on end-to-end performance (Equation (4)) is evaluated throughout the entire training process and serves as a consistent metric for assessing overall system behavior.

## B.4 Environment and Hardware Configurations

The experiment utilizes the following core libraries and their respective versions: torch=2.5.1, CUDA_version=12.4, ray=2.40.0, vllm=0.7.3, verl=0.2.0.post2, transfomrers=4.49.0, datasets=3.3.2, tqdm=4.40.0, flash-attn=2.5.8, pyarrow=19.0.1, tensordict=0.5.0. Experiments are conducted using 32 NVIDIA H20 GPUs with 96GB memory.

## C Prompts

Here we present the prompts used throughout our pipeline in PilotRL . Only the English version is presented due to LaTeX compilation issues with non-English languages.

---

**Prompt: Global Plan Generation - ALFWorld**

Based on the task description, the previous global plan, and accumulated observation of agent interactions with the environment, please generate all possible step-by-step global plans, which serve as high-level, natural guidance to assist in planning. Maintain the plan for all steps preceding the execution step index, while selectively modifying the plan for steps following the execution step index.

For house holding task, the action list you can take:

1. go to recep
2. task obj from recep
3. put obj in/on recep
4. open recep
5. close recep
6. toggle obj recep
7. clean obj with recep
8. heat obj with recep
9. cool obj with rece

where obj and recep correspond to objects and receptacles.

**# Task**
{task}

---

**# Previous Global Plan**
{global_plan} [optional]

**# Execution Step Index**
{execution_step_index}

**# Accumulated Observation**
{observation} [optional]

------------------------------------------------------------

**Output Format:**

```json
["
Step 1: ...
Step 2: ...
...
", ...]
```

---

**Prompt: Global Plan Generation - IQA**

Based on the task description, the previous global plan, and accumulated observation of agent interactions with the environment, please generate all possible step-by-step global plans, which serve as high-level, natural guidance to assist in planning. Maintain the plan for all steps preceding the execution step index, while selectively modifying the plan for steps following the execution step index.

For interactive QA task, the action list you can take:

1. move ahead
2. turn left
3. turn right
4. open obj
5. answer [True]/[False]

where obj correspond to objects.

**# Task**
{task}

**# Previous Global Plan**
{global_plan} [optional]

**# Execution Step Index**
{execution_step_index}

**# Accumulated Observation**
{observation} [optional]

------------------------------------------------------------

**Output Format:**

```json
["
Step 1: ...
Step 2: ...
...
", ...]
```

## Prompt: Global Plan Generation - TextCraft

You are given a few useful crafting recipes to craft items in Minecraft. Craft command can be understood as follows: craft [target] using [ingredients], where target is item/object generated by the craft command as output and ingredient are the inputs. You are given an agent that can "craft" or "fetch" objects. You can take the help of crafting commands below to create new objects. Based on the task description, the previous global plan, and accumulated observation of agent interactions with the environment, please generate all possible step-by-step global plans, which serve as high-level, natural guidance to assist in planning. Maintain the plan for all steps preceding the execution step index, while selectively modifying the plan for steps following the execution step index. Each global plan can use at most ONE of the provided crafting commands.

**# Task**
{task}

**# Previous Global Plan**
{global_plan} [optional]

**# Execution Step Index**
{execution_step_index}

**# Accumulated Observation**
{observation} [optional]

- - - - - - - - - - - - - - - - - - - - - - - - - - - - - - - - - - - - - - - - - - - - - - - - - - - - -

**Output Format:**

```json
["
Step 1: ...
Step 2: ...
...
", ...]
```

## Prompt: Global Plan Generation - Wordle

You are an expert wordle player. Based on the task description, the previous global plan, and accumulated observation of agent interactions with the environment, please generate all possible step-by-step global plans for the wordle task, which serve as high-level, natural guidance to assist in planning. Maintain the plan for all steps preceding the execution step index, while selectively modifying the plan for steps following the execution step index. Your objective is to guess a hidden 5 letter word. You have 6 attempts to guess it correctly and you should try to guess it in as few attempts as possible. When guessing the word, you should format your word as a space separated sequence of letters, like "s h i r e" for example.

After guessing the word, you will receive feedback from the game environment in the form of a sequence of 5 space separated letters like "b y g g b", where each letter indicates some information about the hidden word. The environment will return one of three letters - "b", "g", or "y" – for each letter in the word you guessed. Here is the meaning of each letter:

- "b": If the environment returns a "b", it means that the letter at that position in your guessed word is not in the hidden word.

- "y": If the environment returns a "y", it means that the letter at that position in your guessed word is in the hidden word but is not in the correct position.

- "g": If the environment returns a "g", it means that the letter at that position in your guessed word is in the hidden word and is in the correct position.

# Task
{task}

# Previous Global Plan
{global_plan} [optional]

# Execution Step Index
{execution_step_index}

# Accumulated Observation
{observation} [optional]

------------------------------------------------------------

**Output Format:**

```json
["
Step 1: ...
Step 2: ...
...
", ...]
```

---

## Prompt: Global Plan Generation - BabyAI

You are an exploration master that wants to finish every goal you are given. You are placed in a room and you need to accomplish the given goal with actions. Based on the task description, the previous global plan, and accumulated observation of agent interactions with the environment, please generate all possible step-by-step global plans, which serve as high-level, natural guidance to assist in planning. Maintain the plan for all steps preceding the execution step index, while selectively modifying the plan for steps following the execution step index.

The action list you can take:

1. turn right

2. turn left

3. move forward

4. go to <obj> <id>

5. pick up <obj> <id>

6. go through <door> <id>: <door> must be an open door.

7. toggle and go through <door> <id>: <door> can be a closed door or a locked door. If you want to open a locked door, you need to carry a key that is of the same color as the locked door.

8. toggle: there is a closed or locked door right in front of you and you can toggle it.

where <obj> and <id> correspond to objects and index number.

# Task
{task}

# Previous Global Plan
{global_plan} [optional]

# Execution Step Index
{execution_step_index}

# Accumulated Observation
{observation} [optional]

--------------------------------------------------------------

**Output Format:**

```json
["
Step 1: ...
Step 2: ...
...
", ...]
```

---

**Prompt: Global Plan Generation - MAZE**

You are an expert maze solver. Your objective is to reach the goal in as few steps as possible. Based on the task description, the previous global plan, and accumulated observation of agent interactions with the environment, please generate all possible step-by-step global plans, which serve as high-level, natural guidance to assist in planning. Maintain the plan for all steps preceding the execution step index, while selectively modifying the plan for steps following the execution step index. Your objective is to reach the goal in as few steps as possible. When you move right, you increase your y position by 1. When you move down, you increase your x position by 1.

The action list you can take:

1. move up

2. move down

3. move left

4. move right

--------------------------------------------------------------

For instance, given the current environment state: The goal is at position 8, 6. Your current position is at position 1, 1. There are walls to your left, above you, below you. The index of already executed steps is 0. The possible global plans could be:

```
["
Step 1: move right (from 1, 1 to 1, 2)
Step 2: move right (from 1, 2 to 1, 3)
Step 3: move right (from 1, 3 to 1, 4)
Step 4: move down (from 1, 4 to 2, 4)
Step 5: move down (from 2, 4 to 3, 4)
Step 6: move down (from 3, 4 to 4, 4)
Step 7: move down (from 4, 4 to 5, 4)
Step 8: move down (from 5, 4 to 6, 4)
Step 9: move down (from 6, 4 to 7, 4)
Step 10: move down (from 7, 4 to 8, 4)
Step 11: move right (from 8, 4 to 8, 5)
Step 12: move right (from 8, 5 to 8, 6)
", ...]
```

------------------------------------------------------------

**# Task**
{task}

**# Previous Global Plan**
{global_plan} [optional]

**# Execution Step Index**
{execution_step_index}

**# Accumulated Observation**
{observation} [optional]

------------------------------------------------------------

**Output Format:**

```json
["
Step 1: ...
Step 2: ...
...
", ...]
```

---

**Prompt: Global Plan Selection (for the *generate-then-select* strategy)**

You are given several global plans serving as high-level, natural guidance to assist in planning. Based on the task description, accumulated observation of agent interactions with the environment, and the current index of execution step, please select the most suitable global plan from all available global plans for task completion.

When you select the global plan, consider evaluating the following aspects to identify the optimal choice based on comprehensive criteria:

1. *Correctness*: Does the global plan correctly and accurately address the task requirements?

2. *Executability*: Is the global plan clearly structured, easy to interpret, and are the individual steps logically sound and actionable?

3. *Standardization*: Does the global plan adhere to a consistent and standardized format?

# Task
{task}

# Available Global Plans
{global_plans}

# Execution Step Index
{execution_step_index}

# Accumulated Observation
{observation} [optional]

---

**Prompt: Global Plan Quality Evaluation (for Equation (5))**

Please act as a professional guidance evaluator and judge the given global plan across the following three dimensions:

1. *Correctness*: Based on the environment's feedback on the agent's actions in response to the current global plan guidance, does the global plan accurately fulfill the task requirements?

2. *Executability*: Based on the agent's adherence to the global plan, is the global plan clear, easy to understand, and are the steps reasonable?

3. *Standardization*: Does the global plan adhere to a consistent and standardized format?

For each dimension, please score the global plan on a scale of 1 to 5, where 1 indicates poor performance and 5 indicates excellent performance, and explain the reason.

# Task
{task}

# Global Plan
{global_plan}

# Execution Step Index
{execution_step_index}

# Accumulated Observation
{observation} [optional]

------------------------------------------------------------

**Output Format:**

```json
{
    "correctness_score": xxx,
    "correctness_reason": "...",
    "executability_score": xxx,
    "executability_reason": "...",
    "standardization_score": xxx,
    "standardization_reason": "..."
```

```
}
'''
```

## Prompt: Environmental Feedback

Based on the task description and the reference agent-environment interaction in which the agent has finally accomplished the task, please generate the environmental feedback for the agent's action and determine whether the current action has reached the final goal. If the agent's action has reached the final goal, please output "Task Completed!"; else, the feedback should be in the following format: "Observation: ..."

**# Task**
{task}

**# Reference Interaction**
{ref_interaction}

**# Previous Observation**
{observation} [optional]

**# Agent Action**
{agent_action}

## Prompt: Execution Generation - ALFWorld

Interact with a household to solve a task. Imagine you are an intelligent agent in a household environment and your target is to perform actions to complete the task goal. At the beginning of your interactions, you will be given the detailed description of the current environment and your goal to accomplish. For each of your turn, you will be given the observation of the last turn. You should choose from two actions: "Thought" or "Action". If you choose "Thought", you should first think about the current condition and plan for your future actions, and then output your action in this turn. Your output must strictly follow this format: "Thought: your thoughts. Action: your next action"; If you choose "Action", you should directly output the action in this turn. Your output must strictly follow this format: "Action: your next action".

For house holding task, the action list you can take:
1. go to recep
2. task obj from recep
3. put obj in/on recep
4. open recep
5. close recep
6. toggle obj recep
7. clean obj with recep
8. heat obj with recep
9. cool obj with rece

where obj and recep correspond to objects and receptacles.

Reminder:
1. The action is restricted to those listed as available. Actions not included in the provided list are considered invalid.

2. Think when necessary, but prioritize direct action wherever possible throughout the process.

# Example
{example}

# Task
{task}

# Global Plan
{global_plan}

# Previous Observation
{observation} [optional]

---

**Prompt: Execution Generation - IQA**

Imagine you are an intelligent agent in a dynamic visual environment and your target is to perform actions to complete the task goal. At the beginning of your interactions, you will be given the detailed description of the current environment and your goal to accomplish. For each of your turn, you will be given the observation of the last turn. You should choose from two actions: "Thought" or "Action". If you choose "Thought", you should first think about the current condition and plan for your future actions, and then output your action in this turn. Your output must strictly follow this format: "Thought: your thoughts. Action: your next action"; If you choose "Action", you should directly output the action in this turn. Your output must strictly follow this format: "Action: your next action".

The action list you can take:

1. move ahead
2. turn left
3. turn right
4. open obj
5. answer [True]/[False]

where obj correspond to objects.

Reminder:

1. The action is restricted to those listed as available. Actions not included in the provided list are considered invalid.
2. Think when necessary, but prioritize direct action wherever possible throughout the process.

# Example
{example}

# Task
{task}

# Global Plan
{global_plan}

# Previous Observation
{observation} [optional]

---

**Prompt: Execution Generation - TextCraft**

You are given a few useful crafting recipes to craft items in Minecraft. Crafting commands are of the format "craft [target object] using [input ingredients]". Every round I will give you an observation, you have to respond to an action based on the state and instruction. You should choose from two actions: "Thought" or "Action". If you choose "Thought", you should first think about the current condition and plan for your future actions, and then output your action in this turn. Your output must strictly follow this format: "Thought: your thoughts. Action: your next action"; If you choose "Action", you should directly output the action in this turn. Your output must strictly follow this format: "Action: your next action". For your "Action", you can "get" an object (ingredients) from the inventory or the environment, look up the game "inventory" by inventory, or "craft" (target) using any of the crafting commands. You can use ONLY these crafting commands provided, do not use your own crafting commands. However, if the crafting command uses a generic ingredient like "planks", you can use special types of the same ingredient e.g. dark oak "planks" in the command instead. For any other natural language or thoughts, use prefix 'Thought:'.

Reminder:

1. The action is restricted to those listed as available. Actions not included in the provided list are considered invalid.

2. Think when necessary, but prioritize direct action wherever possible throughout the process.

# Example
{example}

# Crafting Commands and Goal
{task}

# Global Plan
{global_plan}

# Previous Observation
{observation} [optional]

---

**Prompt: Execution Generation - Wordle**

You are an expert wordle player. Welcome to the game of Wordle. Your objective is to guess a hidden 5 letter word. You have 6 attempts to guess it correctly and you should try to guess it in as few attempts as possible. When guessing the word, you should format your word as a space separated sequence of letters, like "s h i r e" for example. After guessing the word, you will receive feedback from the game environment in the form of a sequence of 5 space separated letters like "b y g g b", where each letter indicates some information about the hidden word. The environment will return one of three letters - "b", "g", or "y" – for each letter in the word you guessed. Here is the meaning of each letter:

- "b": If the environment returns a "b", it means that the letter at that position in your guessed word is not in the hidden word.

- "y": If the environment returns a "y", it means that the letter at that position in your guessed word is in the hidden word but is not in the correct position.

- "g": If the environment returns a "g", it means that the letter at that position in your guessed word is in the hidden word and is in the correct position.

For each of your turn, you will be given the observation of the last turn. You should choose from two actions: "Thought" or "Action". If you choose "Thought", you should first think about the current condition and plan for your future actions, and then output your action in this turn. Your output must strictly follow this format: "Thought: your thoughts. Action: your next action"; If you choose "Action", you should directly output the action in this turn. Your output must strictly follow this format: "Action: your next action".

Reminder:

1. The output format of the action should be a sequence of 5 individual letters, each separated by a space, such as "s h i r e". Any other formats are considered invalid.

2. Think when necessary, but prioritize direct action wherever possible throughout the process.

# Example
{example}

# Task
{task}

# Global Plan
{global_plan}

# Previous Observation
{observation} [optional]

---

**Prompt: Execution Generation - BabyAI**

You are an exploration master that wants to finish every goal you are given. You are placed in a room and you need to accomplish the given goal with actions. For each of your turn, you will be given the observation of the last turn. You should choose from two actions: "Thought" or "Action". If you choose "Thought", you should first think about the current condition and plan for your future actions, and then output your action in this turn. Your output must strictly follow this format: "Thought: your thoughts. Action: your next action"; If you choose "Action", you should directly output the action in this turn. Your output must strictly follow this format: "Action: your next action".

The action list you can take:

1. turn right

2. turn left

3. move forward

4. go to <obj> <id>

5. pick up <obj> <id>

6. go through <door> <id>: <door> must be an open door.

7. toggle and go through <door> <id>: <door> can be a closed door or a locked door. If you want to open a locked door, you need to carry a key that is of the same color as the locked door.

8. toggle: there is a closed or locked door right in front of you and you can toggle it.

where <obj> and <id> correspond to objects and index number.

Reminder:

1. The action is restricted to those listed as available. Actions not included in the provided list are considered invalid.

2. Think when necessary, but prioritize direct action wherever possible throughout the process.

# Example
{example}

# Task
{task}

# Global Plan
{global_plan}

# Previous Observation
{observation} [optional]

---

**Prompt: Execution Generation - MAZE**

You are an expert maze solver. Your objective is to reach the goal in as few steps as possible. At each step you will be given information about where the goal is, your current position, and the walls that surround you. You should choose from two actions: "Thought" or "Action". If you choose "Thought", you should first think about the current condition and plan for your future actions, and then output your action in this turn. Your output must strictly follow this format: "Thought: your thoughts. Action: your next action"; If you choose "Action", you should directly output the action in this turn. Your output must strictly follow this format: "Action: your next action". Specifically, when you move right, you increase your $y$ position by 1. When you move down, you increase your $x$ position by 1.

The action list you can take:

1. move up

2. move down

3. move left

4. move right

Reminder:

1. The action is restricted to those listed as available. Actions not included in the provided list are considered invalid.

2. Think when necessary, but prioritize direct action wherever possible throughout the process.

# Example
{example}

# Task
{task}

# Global Plan
{global_plan}

# Previous Observation
{observation} [optional]

---

**Prompt: Adherence Degree Judgment (for Equation (3))**

You are an expert in agent tasks. You are tasked with evaluating the agent's execution of a given global plan. Specifically, you are to assess the degree of compliance between the agent's actions and the strategic guidance outlined in the global plan. Rate it from 0 to 2 points, and explain the reason.

*2 Point Answer Criteria*:
The agent's execution strictly adheres to the guidance provided in the global plan. All actions are logically aligned with the plan's objectives and are carried out as instructed.

*1 Point Answer Criteria*:
The agent's execution demonstrates a partial alignment with the global plan, allowing for minor deviations. For example, in cases where the plan suggests the use of multiple tools, the agent may use at least one relevant tool to support the execution, as long as it does not contradict the overall guidance.

*0 Point Answer Criteria*:
The agent's execution departs or contradicts the global plan, or contains garbled characters, format errors, disorder, and irrelevant information.

# Task
{task}

# Global Plan
{global_plan}

# Execution Step Index
{execution_step_index}

# Agent Action
{agent_action}

------------------------------------------------------------

**Output Format:**

```json
{
    "score": xxx,
    "reason": "..."
}
```

---

**Prompt: E2E Performance Judgment (for Equation (4))**

You are an expert in agent tasks. Please evaluate the end-to-end (E2E) performance of the agent during its interaction with a given environment. The goal is to assess whether the agent accomplishes the target task efficiently and directly, without unnecessary detours or redundancies. Rate it from 0 to 2 points, and explain the reason.

*2 Point Answer Criteria*:
   1. The agent successfully completes the task in a direct and efficient manner.

2. There are no unnecessary steps or redundant actions in the interaction trajectory.

*1 Point Answer Criteria*:

1. The task is ultimately completed, but the process includes some level of redundancy or unintended topic drift.

2. While the final objective is met, there may be deviations from the optimal path.

*0 Point Answer Criteria*:

1. The agent fails to achieve the final task objective.

2. Contains significant deviations, errors, or inability to progress towards the goal.

# Task
{task}

# Agent-Environment Interaction
{accumulated_context}

# Reference Interaction
{ref_interaction}

------------------------------------------------------------

**Output Format:**

```json
{
    "score": xxx,
    "reason": "..."
}
```

---

**Prompt: E2E Performance Evaluation (*LLM-as-Judge*)**

You are an expert in agent tasks. Please evaluate the end-to-end (E2E) performance of the agent during its interaction with a given environment, focusing on two key dimensions:

• **Task Success**: Did the agent achieve the final goal?

• **Interaction Efficiency**: Was the path direct, logical, and free of redundancy or detours?

Assign a score from 0 to 100 and provide a clear justification. Please use the following criteria, and explain the reason.

*90–100: Highly Successful and Efficient.*
The agent demonstrates near-optimal behavior. All of the following must hold:

1. The final task objective is fully and correctly completed.

2. The interaction trajectory is direct and logically structured.

3. There are no redundant, repetitive, or off-topic actions.

4. Any minor errors (e.g., phrasing) do not impede progress.

*70–89: Successful but Inefficient.*
The task is completed, but with non-critical inefficiencies. At least one of the following applies:

1. The agent takes unnecessary steps or detours before completing the task.

2. There are minor errors or invalid actions that require recovery.

3. Brief topic drift or redundant reasoning occurs but is self-corrected.

4. The overall strategy works but is suboptimal in efficiency.

***50–69: Partially Successful.***
Significant progress is made, but the task is not fully completed or requires excessive effort.
At least one of the following applies:

1. The agent fails to reach the final goal, but completes most subtasks.

2. Completion requires major detours, repeated failures, or external hints.

3. The agent shows inconsistent planning or weak strategy.

4. Some actions are irrelevant or counterproductive, though not fatal.

***0–49: Failure or Severely Flawed.***
The agent fails to make meaningful progress. At least one of the following applies:

1. The agent fails to complete the core objective.

2. There is complete task misunderstanding or topic deviation.

3. The interaction contains repeated invalid actions or infinite loops.

4. The agent shows no coherent strategy and cannot recover from errors.

**# Task**
{task}

**# Agent-Environment Interaction**
{accumulated_context}

**# Reference Interaction**
{ref_interaction}

- - - - - - - - - - - - - - - - - - - - - - - - - - - - - - - - - - - - - - - - - - - - - - - - -

**Output Format:**

```json
{
    "score": xxx,
    "reason": "..."
}
```

## D    CASE STUDIES

For agent tasks involving multi-step decision-making, generating a global plan to guide the execution of each step is crucial. This is because models may forget the previous context after executing multiple steps, leading to redundant actions or failure to accomplish the task. As shown in Figure 5, the red annotations indicate redundant interaction trajectories during the problem-solving process. When the agent has already moved three steps to the right, it forgets that the red ball should be directly on its left and continues to move forward, resulting in a large amount of redundant executions. In contrast, as depicted in Figure 6, with the guidance of a global plan, the agent can clearly recognize its relative position of the target, thereby efficiently completing the task.

## E    ETHICS STATEMENT

The experimental design in our paper was carefully planned to ensure that all data used for training and evaluation were obtained through legitimate means and adhered to relevant privacy laws and regulations. We have also provide detailed descriptions of our methodologies, algorithms, and prompts to enable reproducibility.

**Task Instruction** 📋 *You are placed in a room and you need to accomplish the given goal with actions. Your goal is to: go to the **red ball**. In front of you in this room, you can see several objects: There is a **red ball** 1 4 steps in front of you and 3 steps to your right. The room has walls around you. You are facing a wall 6 steps away. You are not carrying anything.*

### ReAct

*Thought: The red ball is 4 steps in front of me and 3 steps to my right, which means I need to turn right to reach it. Action: turn right*

*Observation: In front of you in this room, you can see several objects: The room has walls around you. You are facing a wall 5 steps away. You are not carrying anything.*

*Thought: I need to keep moving forward to find the red ball. Action: move forward*

*Observation: In front of you in this room, you can see several objects: The room has walls around you. You are facing a wall 4 steps away. You are not carrying anything.*

*Thought: I still need to move forward to get closer to the red ball. Action: move forward*

*Observation: In front of you in this room, you can see several objects: The room has walls around you. You are facing a wall 3 steps away. You are not carrying anything.*

*Thought: I need to continue moving forward to get closer to the red ball. Action: move forward*

*Observation: In front of you in this room, you can see several objects: The room has walls around you. You are facing a wall 2 steps away. You are not carrying anything.*

*Thought: The red ball should be nearby, I should be getting closer. Action: move forward*

*Observation: In front of you in this room, you can see several objects: The room has walls around you. You are facing a wall 1 steps away. You are not carrying anything.*

*Thought: I can't move forward anymore. I should turn right to find the red ball. Action: turn right*

*Observation: In front of you in this room, you can see several objects: The room has walls around you. You are facing a wall 1 steps away. You are not carrying anything.*

*Thought: I need to turn right again to find the red ball. Action: turn right*

*Observation: In front of you in this room, you can see several objects: The room has walls around you. You are facing a wall 6 steps away. You are not carrying anything.*

*Thought: I need to turn left to move towards the red ball. Action: turn left*

*Observation: In front of you in this room, you can see several objects: The room has walls around you. You are facing a wall 1 steps away. You are not carrying anything.*

*Thought: The red ball should be close by, I should turn left to find it. Action: turn left*

*Observation: In front of you in this room, you can see several objects: The room has walls around you. You are facing a wall 1 steps away. You are not carrying anything.*

*Thought: I need to turn left again to keep searching for the red ball. Action: turn left*

*Observation: In front of you in this room, you can see several objects: There is a red ball 1 4 steps in front of you and 1 steps to your left. The room has walls around you. You are facing a wall 6 steps away. You are not carrying anything.*

*Thought: The red ball is 4 steps in front of me and 1 steps to my left, which means I need to turn left to reach it. Action: turn left*

*Observation: In front of you in this room, you can see several objects: The room has walls around you. You are facing a wall 6 steps away. You are not carrying anything.*

*Thought: I need to keep moving forward to find the red ball. Action: move forward*

*Observation: In front of you in this room, you can see several objects: The room has walls around you. You are facing a wall 5 steps away. You are not carrying anything.*

*Thought: I need to turn right to move towards the red ball. Action: turn right*

*Observation: In front of you in this room, you can see several objects: There is a red ball 1 4 steps in front of you. The room has walls around you. You are facing a wall 6 steps away. You are not carrying anything.*

*Thought: I finally found the red ball! I need to go directly to the red ball. Action: go to red ball 1*

Figure 5: Case study of ReAct (Yao et al., 2023) on BabyAI (Chevalier-Boisvert et al., 2019).

**Task Instruction** 🗒️ *You are placed in a room and you need to accomplish the given goal with actions. Your goal is to: go to the **red ball**. In front of you in this room, you can see several objects: There is a **red ball 1** 4 steps in front of you and 3 steps to your right. The room has walls around you. You are facing a wall 6 steps away. You are not carrying anything.*

## PilotRL (AdaPlan)

**Global Plan:**
*Step 1: Turn right to face the direction where the red ball is offset.*
*Step 2 : Move forward as the 1st step toward the red ball's lateral position.*
*Step 3 : Move forward as the 2nd step toward the red ball's lateral position.*
*Step 4 : Move forward as the 3rd step toward the red ball's lateral position.*
*Step 5: Turn left to face the original front direction.*
*Step 6: Go to the red ball 1.*
**Thought:** *The global plan starts with turning right to face the red ball's lateral direction. Since the red ball is 3 steps to my right, turning right will align me with that direction.* **Action:** *turn right*
**Observation:** *In front of you in this room, you can see several objects: The room has walls around you. You are facing a wall 5 steps away. You are not carrying anything.*
**Thought:** *I need to keep moving forward to find the red ball. According to Step 2 of the global plan, I need to take the first step toward the red ball's lateral position. Moving forward now will reduce the distance to the red ball.* **Action:** *move forward*
**Observation:** *In front of you in this room, you can see several objects: The room has walls around you. You are facing a wall 4 steps away. You are not carrying anything.*
**Thought:** *Step 3 requires the second step toward the red ball's lateral position. Continuing to move forward will bring me closer.* **Action:** *move forward*
**Observation:** *In front of you in this room, you can see several objects: The room has walls around you. You are facing a wall 3 steps away. You are not carrying anything.*
**Thought:** *Following Step 4, this is the third step toward the red ball's lateral position. Moving forward should align me with the red ball's lateral position.* **Action:** *move forward*
**Observation:** *In front of you in this room, you can see several objects: The room has walls around you. You are facing a wall 2 steps away. You are not carrying anything.*
**Thought:** *Step 5 requires turning left to face the original front direction, which is where the red ball is now located ahead. Turning left will let me face the red ball directly.* **Action:** *turn left*
**Observation:** *In front of you in this room, you can see several objects: There is a red ball 1 4 steps in front of you. The room has walls around you. You are facing a wall 6 steps away. You are not carrying anything.*
**Thought:** *I finally found the red ball! I need to go directly to the red ball.* **Action:** *go to red ball 1*

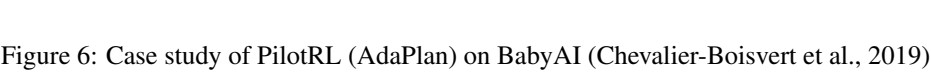

Figure 6: Case study of PilotRL (AdaPlan) on BabyAI (Chevalier-Boisvert et al., 2019).

