# OpenReview forum: "PilotRL: Training Language Model Agents via Global Planning-Guided Progressive Reinforcement Learning"
_ICLR.cc/2026/Conference — ICLR 2026 Conference Withdrawn Submission_

### Official Review · Reviewer_JCPE · 2025-10-23

**Soundness:** 2
**Presentation:** 3
**Contribution:** 2
**Rating:** 4
**Confidence:** 3

**Summary:**

The article proposes a hierarchical learning method for agentic LLM, which features an adaptive global plan-based agent paradigm AdaPlan, and PilotRL, a global planning-guided training framework for LLM agents driven by progressive reinforcement learning. The author does extensive experiments to verify the effectiveness of PilotRL.

**Strengths:**

1. The author does extensive experiments and validates the effectiveness of their method.
2. Despite lack of novelty, the author does provide one possible method to improve the capability of agentic LLM.

**Weaknesses:**

The method lacks novelty. The idea of hierarchical planning has existed for years. Apart from designing a special prompt to extract human rewards, I don't see any prominent innovation. In short, I see too many manually designed components in this paper.

**Questions:**

The selection of plan according to scores from an external expert LLM is too direct. Can it be replaced with an autonomous selection policy trained with RL?

---

> ### Author Response · Authors · 2025-11-18
> **Reply to Reviewer JCPE**
>
> We sincerely appreciate the reviewer's efforts and insightful comments to improve our manuscript. Below, we address the weaknesses (W) and questions (Q) raised.
>
> ---
>
> **[W1] Lack of Novelty**
>
> While it is true that hierarchical planning as a general concept has existed for years, our contribution lies *not in the abstract idea itself*. Below we clarify the key innovations that go beyond prior work:
>
> **1. AdaPlan: A Unified, Adaptive Global Plan-Based Agent Paradigm**
>
> As stated in *line 48–52* (***C1***) of our manuscript, the majority of existing work adopts the ReAct agent paradigm [1], which relies on single-step planning. Our approach introduces a novel agent problem-solving paradigm as a key innovation.
>
> Furthermore, contrary to traditional hierarchical approaches that decouple planner and executor, AdaPlan integrates both the global planner and the executor into a single, end-to-end trainable model. This enables:
> - **Dynamic Plan Refinement**: The global plan is not static; it is continuously updated based on execution feedback.
> - **Synergistic Planning-Execution**: High-level planning directly informs low-level actions, and execution outcomes recursively shape future planning—something rigid two-stage pipelines cannot achieve.
>
> This tight coupling is non-trivial and represents a paradigm shift from conventional hierarchical RL or modular agent designs. As stated in the paper (in *line 88-90*): “*By integrating both the global planner and executor in a unified model, our approach enables more effective coordination and improved end-to-end performance.*”
>
> **2. PilotRL: A Progressive, Global Planning-Guided Reinforcement Learning Framework**
>
> PilotRL introduces a three-stage progressive RL curriculum specifically designed to resolve the tension between planning and execution:
> - Stage 1: Warm-up with supervised plan following.
> - Stage 2: Optimization of global plan generation.
> - Stage 3: End-to-end performance enhancement .
>
> Critically, this staged design addresses a fundamental challenge highlighted in our analysis in *line 356-369* of the manuscript: “*A primary cause of performance drop lies in the intrinsic complexity and potential conflicts among heterogeneous reward signals… conflicting gradient signals… reduce learning efficiency.*”
>
> Our framework systematically mitigates this instability, which prior hierarchical methods do not handle. Thus, the “manual design” is not arbitrary—it is a principled solution to a newly identified training pathology in LLM-based agents.
>
> In conclusion, our work represents a meaningful architectural (**AdaPlan**) and algorithmic advance (**PilotRL**), not a mere repackaging of existing ideas. We hope this clarifies the substantive novelty of our paper.
>
> ---
>
> **[Q1] Plan Selection by an External Expert LLM**
>
> We thank the reviewer for this insightful question. We adapt the strategy of selecting plans via an external expert LLM since it is advantageous in the context of natural language planning, for the following reasons:
>
> 1. **Robustness with Deep Semantic Understanding**: Global plans in our framework are *free-form natural language sequences*. Traditional rule-based or syntactic filters (e.g., regex or grammar parsers) are notoriously brittle under linguistic variation. In contrast, the external expert LLM leverages deep semantic understanding and exhibits strong robustness to surface-form variation, enabling more reliable assessment of plan correctness, executability and format validity.
>
> 2. **Providing a Stable, High-Quality Prior**: While an RL-trained autonomous selector is theoretically possible, plan selection over natural language candidates is a high-dimensional, discrete, and sparsely rewarded decision problem. *In early training stages, an RL policy would lack sufficient signal to distinguish good from bad plans and likely converge to degenerate solutions.* In contrast, the external expert LLM can provide a stable, high-quality prior that guides the agent toward semantically coherent planning behavior, which serves as a crucial scaffold for subsequent training stages.
>
> Thus, rather than being “too direct”, our design reflects an empirical approach to balancing expressivity, robustness, and trainability in natural language agents.
>
> ---
>
> > [1] Yao S, Zhao J, Yu D, et al. React: Synergizing reasoning and acting in language models[C]//The eleventh international conference on learning representations. 2022.

---

> ### Author Response · Authors · 2025-11-28
>
> We sincerely hope these detailed clarifications have addressed your concerns comprehensively. We are deeply grateful for your thorough and constructive feedback, which has significantly helped us improve the clarity and rigor of our paper. If our rebuttal has addressed your concerns satisfactorily, we would be honored if you could consider reflecting this in your final score.

---

> ### Author Response · Authors · 2025-11-28
>
> Thank you for your thoughtful and constructive feedback on our work. We have carefully addressed your comments and revised the work and manuscript accordingly. *As the discussion period nears its end, we would greatly appreciate any additional questions or points of clarification.* If our responses have satisfactorily addressed your concerns, we kindly ask you to consider reflecting this in your score.
>
> Thanks again for your time and expertise.

---

### Official Review · Reviewer_juHn · 2025-10-28

**Soundness:** 3
**Presentation:** 3
**Contribution:** 3
**Rating:** 6
**Confidence:** 3

**Summary:**

This paper addresses key limitations of existing Large Language Model (LLM)-based agents in complex, long-horizon tasks, including: (1) the ReAct paradigm’s short-sightedness (focused on single-step reasoning rather than long-term planning); (2) poor coordination between isolated planners and executors; and (3) weak generalization of Supervised Fine-Tuning (SFT), which leads models to memorize task trajectories instead of learning transferable skills.
To solve these issues, the authors propose two core components:
AdaPlan: An adaptive global planning paradigm that integrates a global planner and executor into a unified model. The planner dynamically generates and updates high-level global plans based on real-time environmental feedback, guiding the executor to avoid short-sighted decisions and align actions with long-term goals.
PilotRL: A global planning-guided progressive reinforcement learning (RL) framework built on AdaPlan. It trains LLMs in three sequential stages: (1) enhancing the executor’s ability to follow global plans; (2) optimizing the quality of plans generated by the planner; (3) jointly refining the coordination between planning and execution.
Extensive experiments on 6 benchmarks (e.g., ALFWorld, BabyAI) show that PilotRL achieves state-of-the-art performance. For example, LLaMA3.1-8B-Instruct (an open-source model) trained with PilotRL outperforms the closed-source GPT-4o by 3.60% and GPT-4o-mini by 55.78% (at comparable parameter scales), and exhibits strong generalization on out-of-domain tasks.

**Strengths:**

1. AdaPlan introduces an adaptive global planning mechanism that dynamically generates and updates high-level plans based on real-time environmental feedback. Critically, it unifies the global planner and executor into a single model, solving the "isolation problem" of prior work (where planners and executors were trained separately, leading to misalignment.
2.  Instead of using naive RL or SFT, PilotRL’s three-stage training (executor enhancement → planner optimization → joint coordination) is a novel combination of skill-building and RL. This sequential design avoids conflicting training signals (e.g., optimizing plan quality before the executor can follow plans) and addresses SFT’s generalization flaw by using RL to foster transferable skills rather than trajectory memorization.

**Weaknesses:**

1. The paper’s methodology is heavily dependent on DeepSeek-V3 for two critical roles: (1) generating initial global plans and (2) evaluating key metrics. It does not test whether replacing DeepSeek-V3 with other models (e.g., open-source alternatives like LLaMA3.1-70B-Instruct or closed-source GPT-4o) would preserve PilotRL’s performance. This makes it unclear if PilotRL’s success is inherent to its design or contingent on DeepSeek-V3’s quality.
2. The paper evaluates PilotRL on 6 benchmarks, but all tasks are text-only or lightweight symbolic environments (e.g., ALFWorld: text-based household navigation; Wordle: word guessing; BabyAI: grid-world navigation). It fails to test PilotRL on scenarios that reflect real-world agent challenges.
3. The paper emphasizes PilotRL’s successes (e.g., outperforming GPT-4o; efficient BabyAI navigation) but provides almost no analysis of when and why PilotRL fails. For example: Does PilotRL struggle with tasks requiring extreme long-horizon planning (e.g., 20+ steps vs. 5–10 steps in current benchmarks)? How does it perform when environmental feedback is noisy or delayed (e.g., "observation arrives 3 steps after action")?

**Questions:**

See the Weaknesses Section.

---

> ### Author Response · Authors · 2025-11-20
> **Reply to Reviewer juHn on W1 & W2**
>
> We sincerely appreciate the reviewer's efforts and insightful comments to improve our manuscript. Below, we address the weaknesses (W) and questions (Q) raised.
>
> ---
>
> **[W1] Alternatives of DeepSeek-V3**
>
> Thank you for your suggestion. Here we use **Qwen2.5-7B-Instruct** as the **backbone model** and conduct experiments by replacing DeepSeek-V3 with LLaMA3.1-70B-Instruct and GPT-4o. The results are summarized in the table below. The best and second best of each model are in **bold** and *italic*.
>
> | Model for Evaluation | Method | w/o Plan. | ALFWorld | IQA | TextCraft | Wordle | BabyAI | MAZE | Avg. |
> |-|-|-|-|-|-|-|-|-|-|
> | LLaMA3.1-70B-Instruct | Naive Response  | ✗ | 45.92 | 32.18 | 27.61 | 31.45  | 37.63  | 30.54 | 34.22 |
> | | ReAct  | ✗  | 49.33    | 34.82 | 31.24   | 37.68  | 40.87  | 34.71 | 38.11 |
> | | +MPO  | ✓  | 64.57    | 55.39 | 49.52   | 53.41  | 51.12  | 46.43 | 53.41 |
> | | SFT  | ✓  | *64.78*    | 60.12 | *70.34*  | *71.89*  | 52.45  | 44.15 | *60.62* |
> | | Vanilla RL  | ✗   | 62.73    | *61.54* | 67.98   | 68.02  | *55.89*  | *47.31* | 60.58 |
> | | **PilotRL (ours)** | ✓  | **68.05**  | **64.61** | **72.63**   | **74.42**  | **58.79**  | **54.67** | **65.53** |
> | GPT-4o | Naive Response  | ✗  | 51.94  | 36.15 | 33.62   | 35.98  | 43.62  | 34.57 | 39.31 |
> | | ReAct  | ✗  | 53.38    | 38.32 | 35.71  | 41.19  | 45.31  | 38.26 | 42.03 |
> | | +MPO | ✓  | 68.56    | 59.87 | 53.05   | 58.03  | 54.63  | 50.92 | 57.51 |
> | | SFT | ✓  | *68.79*    | 64.12 | *74.36*   | *75.88*  | 56.43  | 48.17 | 64.63 |
> | | Vanilla RL | ✗ | 66.72    | *65.53* | 71.94    | 72.51  | *59.87*  | *51.34* | *64.65* |
> | | **PilotRL (ours)** | ✓ | **72.03**    | **68.62** | **76.61**   | **78.45**  | **62.81**  | **58.76** | **69.55** |
>
> It can be observed that, due to differences in model scoring preferences, there is indeed some variation in scores under the LLM-as-Judge paradigm. Nevertheless, our PilotRL consistently outperforms the other baselines overall.
>
> ---
>
> **[W2] Scenarios of Real-World Agent Challenges**
>
> We appreciate the reviewer’s comment regarding real-world applicability. We select the benchmarks in our experiments  for several reasons:
>
> 1. **Widely Accepted Community Standard**: These environments are widely adopted as *standard testbeds* in the LLM agent literature[1][2][3], which jointly evaluate key agent capabilities, including long-horizon planning, partial observability, tool grounding, and error recovery. Therefore, these benchmarks can effectively validate that our PilotRL addresses *fundamental algorithmic gaps*.
>
> 2. **A Foundation for Complex Agents**: We view this work as *a foundational step toward more complex agents*. Just as *ReAct* [4] was first validated in symbolic environments before being deployed in real-world systems, *PilotRL* establishes a robust training paradigm for adaptive planning under sparse rewards, which is a prerequisite for scaling to realistic scenarios.
>
> We fully agree that future work should evaluate PilotRL on multimodal as well as more complex platforms, and we plan to pursue this direction.
>
> ---
>
> > [1] Song Y, Xiong W, Zhao X, et al. AgentBank: Towards Generalized LLM Agents via Fine-Tuning on 50000+ Interaction Trajectories[C]//Findings of the Association for Computational Linguistics: EMNLP 2024. 2024: 2124-2141.
>
> > [2] Song Y, Xiong W, Zhao X, et al. AgentBank: Towards Generalized LLM Agents via Fine-Tuning on 50000+ Interaction Trajectories[C]//Findings of the Association for Computational Linguistics: EMNLP 2024. 2024: 2124-2141.
>
> > [3] Liu X, Yu H, Zhang H, et al. AgentBench: Evaluating LLMs as Agents[C]//The Twelfth International Conference on Learning Representations. 2023.
>
> > [4] Yao S, Zhao J, Yu D, et al. React: Synergizing reasoning and acting in language models[C]//The eleventh international conference on learning representations. 2022.

---

> ### Author Response · Authors · 2025-11-20
> **Reply to Reviewer juHn on W3**
>
> **[W3] Analysis of PilotRL's Failure**
>
> We thank the reviewer for this constructive feedback. Below, we address the two concerns raised.
>
> 1. **Environmental Feedback Latency**: In all our experiments, environmental observations are provided in real-time. For example, in *ALFWorld*, the agent executes an action (e.g., "`take key`") and immediately receives the environment’s response (e.g., "`You pick up the key`"). Similarly, in *Wordle*, the agent guesses a word and instantly receives color-coded feedback for each letter. This design ensures no delayed feedback. However, we acknowledge that testing under delayed or noisy feedback would be a valuable extension, which we plan to explore in future work.
>
> 2. **Failure Cases in Complex Tasks**: We manually inspected PilotRL’s failures and found that ***most errors occur in tasks requiring intricate planning or ambiguous reasoning***. For example:
>    - The agent struggled with tasks requiring multi-step text editing (e.g., "`replace 'cat' with 'dog' in the third paragraph`"). The plan lacked specificity (e.g., "`edit text`"), and the final action failed to locate the correct context.
>    - The agent misanswered a question like "`What is the capital of a country whose flag has a lion and a star?`" due to ambiguous clues (e.g., "`lion and star`" could refer to multiple countries).
>
>     As shown in *Figure 3* of the paper, while the global plan generation capability stabilizes during the later training stages, the final performance is still *suboptimal* due to such edge cases, and the agent may struggle when the task exceeds its learned patterns or task definitions are ambiguous.

---

> ### Author Response · Authors · 2025-11-28
>
> We sincerely hope these detailed clarifications have addressed your concerns comprehensively. We are deeply grateful for your thorough and constructive feedback, which has significantly helped us improve the clarity and rigor of our paper. If our rebuttal has addressed your concerns satisfactorily, we would be honored if you could consider reflecting this in your final score.

---

> ### Author Response · Authors · 2025-11-28
>
> Thank you for your thoughtful and constructive feedback on our work. We have carefully addressed your comments and revised the work and manuscript accordingly. *As the discussion period nears its end, we would greatly appreciate any additional questions or points of clarification.* If our responses have satisfactorily addressed your concerns, we kindly ask you to consider reflecting this in your score.
>
> Thanks again for your time and expertise.

---

### Official Review · Reviewer_WDTv · 2025-10-31

**Soundness:** 2
**Presentation:** 4
**Contribution:** 1
**Rating:** 2
**Confidence:** 3

**Summary:**

This paper introduces ​​PILOTRL​​, a training framework for LLM agents that leverages global planning-guided progressive reinforcement learning. It proposes the ​​AdaPlan paradigm​​, which synergizes high-level explicit guidance with execution through dynamic plan adaptation, enabling long-horizon decision-making in complex tasks. It also proposed a method for fine-tuning using reinforcement learning (RL) to enhance the agent's capabilities.

**Strengths:**

- The AdaPlan architecture dynamically updates global plans based on real-time environmental feedback, allowing agents to adjust strategies mid-execution.

- PILOTRL employs a three-stage reinforcement learning pipeline that incrementally develops agent capabilities. This progressive approach mitigates the pitfalls of single-step paradigms and enhances generalization, as evidenced by robust performance on both in-domain and out-of-domain tasks.

**Weaknesses:**

- All reward functions in the paper are implemented using DeepSeek-V3. This raises a contradictory issue: Is DeepSeek-V3's reward evaluation accurate? If DeepSeek-V3 can correctly assess whether a task is completed, it implies that it fully understands how the task should be correctly accomplished, and furthermore, it should have the capability of understanding the PilotRL workflow. Theoretically, DeepSeek-V3 could independently complete the task on its own, without needing the PilotRL fine-tuning process.

- The paper does not compare against agents based on DeepSeek-V3.

- The core idea of AdaPlan proposed in the paper is to dynamically adjust the planning content in real-time based on execution status. This is quite simple and has limited novelty [1] [2].

- The fundamental challenge for RL-based LLM Agents lies in accurately constructing the reward function. The paper does not address this core issue. This work is more like an A+B combination of "LLM Agent Planning" techniques and "LLM Agent RL based on LLM reward" techniques.

[1] Huang, Xu, et al. "Understanding the planning of LLM agents: A survey." arXiv preprint arXiv:2402.02716 (2024).

[2] Jia, Ziqi, et al. "Enhancing Multi-Agent Systems via Reinforcement Learning with LLM-based Planner and Graph-based Policy." arXiv preprint arXiv:2503.10049 (2025).

**Questions:**

- The paper uses DeepSeek-V3 as the reward function. Could you provide the accuracy rate of DeepSeek-V3 in completing the tasks described in the paper?

- If DeepSeek-V3 cannot provide correct rewards, how does the PilotRL proposed in the paper address this issue?

---

> ### Author Response · Authors · 2025-11-19
> **Reply to Reviewer WDTv on W1 & W2 &Q1**
>
> We sincerely appreciate the reviewer's efforts and insightful comments to improve our manuscript. Below, we address the weaknesses (W) and questions (Q) raised.
>
> ---
>
> **[W1] Concerns on Using DeepSeek-V3 for Reward Evaluation**
>
> We appreciate the reviewer’s thoughtful concern. However, there is no contradiction in using DeepSeek-V3 as a reward model while still requiring PilotRL training for task execution for the following reasons.
>
> 1. **Evaluation ≠ Execution**: Accurate evaluative capability does not necessarily entail effective execution ability. This stems from a well-documented asymmetry in large language models: ***the ability to evaluate a solution is often significantly easier than the ability to generate it***—a phenomenon observed across multiple decision-making tasks [1][2].
> 2. **Reference Interaction Trajectories**: Moreover, when DeepSeek-V3 evaluates task completion, we additionally provide reference trajectories—labeled as `# Reference Interaction` in *line 1611-1648*. These reference trajectories are sourced from [3][4], as noted in *line 805-807* of the paper. They further supply DeepSeek-V3 with a concrete reference standard to guide and calibrate its reward judgments, including task completion and solution efficiency.
>
> 3. **Efficiency of LLM-as-Judge**: Such evaluation strategy is essentially an LLM-as-Judge approach (in *line 246-248*), which has become a standard evaluation approach for LLM agents, as human evaluation is costly and unscalable for long-horizon, open-ended tasks. This paradigm is now widely adopted in major agent benchmarks where rule-based metrics fail to capture semantic correctness[5][6].
>
> ---
>
> **[W2] && [Q1] Comparison against Agents based on DeepSeek-V3**
>
> Thank you for your suggestion. In our evaluation, we include **DeepSeek-V3** as one of the baselines under the category of ***Close-Sourced Models***. To ensure a fair comparison, we assess its end-to-end (E2E) performance using the same criteria applied to other Close-Sourced Models—such as GPT-4o and GPT-4o-mini, whose results are reported in *line 277-278* of *Table 1*. Specifically, we jointly consider both task completion rate and efficiency of the interaction trajectory, following the evaluation protocol detailed in *line 246-248* of our paper.
>
> | Backbone Model | Method | w/o Plan. | ALFWorld | IQA | TextCraft | Wordle | BabyAI | MAZE | Avg. |
> |-|-|-|-|-|-|-|-|-|-|
> | DeepSeek-V3 | –  | ✗ | 78.62 | 65.35 | 70.15 | 80.78 | 59.35 | 62.58 | 69.47 |
>
> As shown in the results, DeepSeek-V3 achieves an average score of 69.47 under the evaluation criteria. Although the score is not particularly high, it remains fully qualified to serve as an evaluator for assessing model performance. This is grounded in the statement presented in our response to **[W1]**, which emphasizes the fundamental distinction between evaluation and execution (i.e., Evaluation ≠ Execution).
>
> ---
>
> > [1] Madaan A, Tandon N, Gupta P, et al. Self-refine: Iterative refinement with self-feedback[J]. Advances in Neural Information Processing Systems, 2023, 36: 46534-46594.
>
> > [2] Lightman H, Kosaraju V, Burda Y, et al. Let's verify step by step[C]//The Twelfth International Conference on Learning Representations. 2023.
>
> > [3] Song Y, Xiong W, Zhao X, et al. AgentBank: Towards Generalized LLM Agents via Fine-Tuning on 50000+ Interaction Trajectories[C]//Findings of the Association for Computational Linguistics: EMNLP 2024. 2024: 2124-2141.
>
> > [4] Xi Z, Ding Y, Chen W, et al. Agentgym: Evolving large language model-based agents across diverse environments[J]. arXiv preprint arXiv:2406.04151, 2024.
>
> > [5] Zheng L, Chiang W L, Sheng Y, et al. Judging llm-as-a-judge with mt-bench and chatbot arena[J]. Advances in neural information processing systems, 2023, 36: 46595-46623.
>
> > [6] Li D, Jiang B, Huang L, et al. From generation to judgment: Opportunities and challenges of llm-as-a-judge[C]//Proceedings of the 2025 Conference on Empirical Methods in Natural Language Processing. 2025: 2757-2791.

---

> ### Author Response · Authors · 2025-11-19
> **Reply to Reviewer WDTv on W3 & W4 &Q2**
>
> **[W3] Novelty of AdaPlan**
>
> Thanks for your suggestions. While dynamic planning as a general concept has been discussed in previous works, ***AdaPlan introduces a concrete, end-to-end trainable architecture that unifies global planning and local execution within a single LLM***, enabling real-time plan refinement through environmental feedback. Prior cited works[1][2] fall short in key aspects:
>
> - Existing agent paradigms mentioned in [1], e.g., ReAct, Plan-and-Execute, are lack of contextual awareness and *poor planner-executor coordination*, which are precisely the limitations AdaPlan addresses.
> - Jia et al.[2] employs a *separate* LLM-based planner and graph-based executor, resulting in a *modular, non-adaptive* pipeline where the plan is fixed after generation. In contrast, AdaPlan’s plan is continuously updated during execution, forming a closed-loop reasoning process.
>
> ---
>
> **[W4] Challenge of Constructing the Reward Function**
>
> We thank the reviewer for this insightful question. However, PilotRL is explicitly designed to address the very core challenge the reviewer identifies: the difficulty of constructing effective reward signals for RL-based LLM agents.
>
> Our key innovation lies in the following aspects:
>
> **1. Unified Planner-Executor Architecture**: Unlike prior approaches that decouple planning and execution into separate modules, we integrate both functions into a single, end-to-end trainable LLM. This unification ensures that *the same internal representation* is used for both planning and action, eliminating misalignment between planner and executor that often corrupts reward signals in modular designs. Experimental results and analysis supporting this statement are provided in *line 416-431*, *Table 4* and *Table 8*.
>
> **2. Progressive Curriculum**: Recognizing that naive E2E RL fails due to sparse and conflicting rewards, we design a three-stage training curriculum，each stage equipped with tailored reward functions that match the learning capability of the agent at that phase.
> - Stage 1: Warm-up with supervised plan following.
> - Stage 2: Optimization of global plan generation.
> - Stage 3: End-to-end performance enhancement .
>
> Ablation studies (*Section 4.1*, *Table 2* and *Table 6*) confirm that skipping any stage or using a single reward scheme throughout leads to performance degradation, underscoring the necessity of our adaptive reward design.
>
> ---
>
> **[Q2] Concerns on Reward Correctness**
>
> We thank the reviewer for raising this important concern. While it is true that LLM-as-Judge (here we employ DeepSeek-V3 as our evaluator) may occasionally assign incorrect rewards on individual trajectories, our training framework is specifically designed to be robust to such imperfections.
>
> **Advantages of GRPO.** We employ GRPO as our RL algorithm, which learns from *comparisons among trajectory groups* rather than absolute reward magnitudes. This makes the learning process highly tolerant to occasional reward noise: as long as the reward model preserves a reasonable ranking (i.e., good trajectories tend to receive higher scores than poor ones on average), GRPO can still extract meaningful signal. In this case, perfect reward accuracy is not required, only approximate discriminability.
>
> ---
>
> > [1] Huang, Xu, et al. "Understanding the planning of LLM agents: A survey." arXiv preprint arXiv:2402.02716 (2024).
>
> > [2] Jia, Ziqi, et al. "Enhancing Multi-Agent Systems via Reinforcement Learning with LLM-based Planner and Graph-based Policy." arXiv preprint arXiv:2503.10049 (2025).

---

> ### Author Response · Authors · 2025-11-28
>
> We sincerely hope these detailed clarifications have addressed your concerns comprehensively. We are deeply grateful for your thorough and constructive feedback, which has significantly helped us improve the clarity and rigor of our paper. If our rebuttal has addressed your concerns satisfactorily, we would be honored if you could consider reflecting this in your final score.

---

> ### Author Response · Authors · 2025-11-28
>
> Thank you for your thoughtful and constructive feedback on our work. We have carefully addressed your comments and revised the work and manuscript accordingly. *As the discussion period nears its end, we would greatly appreciate any additional questions or points of clarification.* If our responses have satisfactorily addressed your concerns, we kindly ask you to consider reflecting this in your score.
>
> Thanks again for your time and expertise.

---

### Official Review · Reviewer_hZBX · 2025-11-01

**Soundness:** 3
**Presentation:** 2
**Contribution:** 3
**Rating:** 6
**Confidence:** 3

**Summary:**

This paper introduces PILOTRL, a novel training framework for LLM agents, designed to overcome the limitations of existing paradigms. The authors identify three primary challenges with current methods: (1) the ReAct paradigm's focus on single-step reasoning, which is insufficient for long-horizon tasks; (2) poor coordination between isolated planner and executor modules ; and (3) the tendency of Supervised Fine-Tuning (SFT) to memorize trajectories, which limits generalization.

To address this, the authors propose a two-part solution. First, they introduce AdaPlan, an adaptive agent paradigm where a unified model acts as both a global planner and an executor. This global plan is not static but is continuously updated based on environmental feedback. Second, they propose PilotRL, a training framework based on AdaPlan that uses progressive RL. Experiments show that an 8B open-source model (LLaMA3.1) trained with PilotRL achieves state-of-the-art results, surpassing the closed-source GPT-4o by 3.60% and significantly outperforming baselines like SFT and standard RL.

**Strengths:**

The core contribution, the progressive reinforcement learning curriculum, is a significant and novel approach. Instead of tackling the complex, multi-objective problem of planning and acting simultaneously, the framework logically scaffolds the agent's capabilities. It first learns to follow (Stage 1), then to plan (Stage 2), and finally to coordinate (Stage 3). This staged approach is a highly intuitive and effective solution to the inherent difficulty of training complex, multi-faceted agents.

The AdaPlan paradigm itself is a robust architectural choice. By unifying the planner and executor into a single model , the framework directly addresses the "coordination problem"  that plagues isolated, multi-model systems. Furthermore, the plan is adaptive, meaning it is refined based on new observations, which is demonstrably more resilient than "one-shot" planning.

The paper's empirical results are strong. The claim that an 8B open-source model can be trained to surpass GPT-4o on a suite of agentic benchmarks is a major finding. This suggests that advanced, progressive RL training, rather than just model scale or pre-training, may be a key to unlocking high-level agentic capabilities, offering a path for open-source models to compete with proprietary systems.

The paper is clearly written. The authors effectively use diagrams to contrast their approach with the limitations of ReAct, SFT, and isolated planners . The three-stage training process is well-defined, and the motivation for each stage is persuasively argued

**Weaknesses:**

The paper states, "we employ the frontier model DeepSeek-V3 to simulate real-world environmental behaviors". This is a potentially confounding, methodological choice. For tasks like ALFWorld and BabyAI, which have well-defined, executable simulators, the agent is not interacting with the actual environment. Instead, it is interacting with another LLM (DeepSeek-V3) that simulates that environment. This abstraction means the agent is learning to solve a text-based language game with DeepSeek-V3, not necessarily learning to solve the embodied task. This setup undermines the claims of performance on benchmarks like ALFWorld.

In the ablation for unified vs. isolated architectures (Table 4), the isolated model was trained by "separately training the planner and executor modules... with each component trained for 2 epochs" . This does not appear to be an apples-to-apples comparison. The unified model received a 4-epoch progressive curriculum. It's unclear if the isolated model received the same total number of training steps or the same progressive curriculum. The 5.63% performance drop  could be an artifact of a different, and potentially suboptimal, training scheme, rather than a true reflection of the architecture's inferiority

**Questions:**

1. Could the authors please clarify the exact interaction loop for a task like ALFWorld? When the PilotRL agent outputs "Action: go to tustand" , is this action fed into the canonical ALFWorld simulator to get a new state, or is it fed back into DeepSeek-V3 , which then generates a plausible text observation (e.g., "On the tustand, you see a vase...")?

2. The training and evaluation are deeply reliant on DeepSeek-V3 as both a reward model and the final judge. Were any steps taken to mitigate the risk that PilotRL is simply "reward hacking" or "overfitting to the judge"? For instance, was a different high-capability LLM (like GPT-4o) ever used as a "second opinion" judge during evaluation to see if the performance gains hold?

3. To ensure a fair comparison in Table 4, was the "Isolated" model trained with the same progressive 3-stage curriculum as the unified model (e.g., Stage 1 training only the executor, Stage 2 only the planner, Stage 3 both)? If not, isn't it possible the performance gap is due to the lack of progressive training, rather than the isolated architecture itself?

---

> ### Author Response · Authors · 2025-11-18
> **Reply to Reviewer hZBX on W1**
>
> We sincerely appreciate the reviewer's efforts and insightful comments to improve our manuscript. Below, we address the weaknesses (W) and questions (Q) raised.
>
> ---
>
> **[W1] Role of DeepSeek-V3**
>
> We employ DeepSeek-V3 to simulate real-world environmental behaviors for the following reasons:
> 1. **Reliability of DeepSeek-V3**: Inspired by ZeroSearch [1], the frontier models such as DeepSeek-V3 have acquired extensive commonsense knowledge and reasoning capabilities during pretraining through exposure to trillions of high-quality tokens (e.g., understanding that “a remote control should be placed on the coffee table”). This endows it with strong semantic task understanding, enabling it to directly infer both the agent’s current state—such as the spatial configuration and locations of objects—and the desired goal state from the multi-turn dialogue context in benchmarks like ALFWorld and BabyAI. In contrast, *native simulators*, while offering high-fidelity interactive environments, provide only low-level perceptual feedback (e.g., “room layout,” “object positions”) without explicit semantic interpretation of the task objective.
>     - For example, given a conversation:
>     ```
>     "Human": "You are in the middle of a room. Looking quickly around you, you see a bathtubbasin 1, a cabinet 2, a cabinet 1, a countertop 1, a garbagecan 1, a handtowelholder 1, a sinkbasin 1, a toilet 1, a toiletpaperhanger 1, and a towelholder 1. Your task is to: put a toiletpaper in toiletpaperhanger."
>     "gpt": "Action: go to toiletpaperhanger 1",
>     "Human": "Observation: On the toiletpaperhanger 1, you see nothing.",
>     "gpt": "Action: go to toilet 1",
>     "Human": "Observation: On the toilet 1, you see a soapbottle 1, and a toiletpaper 1.",
>     "gpt": "Action: take toiletpaper 1 from toilet 1",
>     "Human": "Observation: You pick up the toiletpaper 1 from the toilet 1.",
>     "gpt": "Action: go to toiletpaperhanger 1",
>     "Human": "Observation: On the toiletpaperhanger 1, you see nothing.",
>     "gpt": "Action: put toiletpaper 1 on toiletpaperhanger 1"
>     ```
>     From this conversation alone, DeepSeek-V3 can immediately deduce that “*there is a toiletpaper on toilet 1*” and that the intended final state is “*toiletpaper placed on toiletpaperhanger 1.*” In contrast, a native simulator would typically require the agent to perform sequential environment queries—such as exhaustively inspecting every object or location—to reconstruct the same information, resulting in slower, less robust, and more error-prone planning.
>
> 2. **Computational Efficiency**: Moreover, DeepSeek-V3 offers significantly lower computational overhead during inference compared to executing interactions in a physical simulator. *Native simulators* like ALFWorld rely on graphics engines such as Unity or AI2-THOR, which necessitate loading 3D scenes, rendering visual inputs, and maintaining complex state machines. This would significantly increase the computational overhead during the rollout phase (simulating multi-turn interactions with the environment) in RL training. By contrast, DeepSeek-V3 operates purely on textual representations and can be deployed efficiently on standard hardware, enabling large-scale experimentation even under constrained computational budgets.
>
> ---
>
> > [1] Sun H, Qiao Z, Guo J, et al. Zerosearch: Incentivize the search capability of llms without searching[J]. arXiv preprint arXiv:2505.04588, 2025.

---

> ### Author Response · Authors · 2025-11-18
> **Reply to Reviewer hZBX on W2 & Q1 & Q2 & Q3**
>
> **[W2 && Q3] Ablation for Unified vs. Isolated Architectures**
>
> Thank you very much for your thoughtful and detailed feedback.
>
> 1. **Definition of “Unified vs. Isolated Architectures”**: The terms “Unified” and “Isolated” refer specifically to the training method. As noted in *line 52–55* (***C2***) of our paper, prior work typically trains the planner and executor separately, which often leads to suboptimal coordination between planning and execution. In our context, a “*Unified Architecture*” denotes a training regime where the model’s planning and execution capabilities are jointly reinforced in an end-to-end manner. In contrast, an “*Isolated Architecture*” refers to training two distinct models: one exclusively optimized for planning (without any explicit objective for execution) [1], and another solely focused on execution (without planning awareness).
>
> 2. **Training Budget Alignment**: In the original PilotRL pipeline, the training schedule consists of (4 epochs in total):
>     - Stage 1 (execution-centric): 1 epoch
>     - Stage 2 (planning-centric): 2 epochs
>     - Stage 3 (E2E training): 1 epoch
>
>     For the ablation study, we trained the isolated planner for 2 epochs and the isolated executor for 2 epochs, resulting in a total training budget of 4 epochs—identical to that of the unified setting. Notably, the isolated executor even received more execution-focused training (2 epochs) than the unified model (1 epoch in Stage 1), further ensuring a conservative and fair comparison.
>
> 3. **On Using the Same Progressive 3-Stage Curriculum as the Unified Model**: Regarding the suggestion to apply the same progressive 3-stage curriculum to each isolated component, we note two key considerations:
>     1. If we were to train both the planner and executor through the full 4-epoch PilotRL curriculum individually, the total training cost would double (8 epochs combined), which would constitute an unfair advantage in computational budget and violate the principle of controlled ablation.
>     2. Such a setup would essentially amount to training two identical instances of the same PilotRL-trained model (e.g., Qwen2.5-7B-Instruct + PilotRL) — one assigned as planner, the other as executor. In fact, this is precisely the strategy we adopt during inference to accelerate benchmark evaluation: we deploy multiple copies of the same PilotRL-trained model in parallel to handle different tasks or roles. Therefore, this configuration does not represent a meaningful architectural ablation but rather a deployment optimization.
>
> ---
>
> **[Q1] Description of the Interaction Loop**
>
> When the PilotRL agent outputs an action such as "Action: go to tvstand," this action is fed back into DeepSeek-V3, which then generates a plausible textual observation. The rationale for this design is detailed in **[W1]**.
>
> ---
>
> **[Q2] Judgement of Evaluation**
>
> We employ human evaluation as a "second-opinion" judge. During training, we logged the original verification data, and for each global batch, we sampled 16 instances. From each instance, we selected 2 rollouts along with their corresponding human verification to assess the reasonableness of the reward model’s judgments.
>
> ---
>
> > [1] Weimin Xiong, Yifan Song, Qingxiu Dong, Bingchan Zhao, Feifan Song, XWang, and Sujian Li. 2025. MPO: Boosting LLM Agents with Meta Plan Optimization. In Findings of the Association for Computational Linguistics: EMNLP 2025, pages 3914–3935, Suzhou, China. Association for Computational Linguistics.

---

> ### Author Response · Authors · 2025-11-28
>
> We sincerely hope these detailed clarifications have addressed your concerns comprehensively. We are deeply grateful for your thorough and constructive feedback, which has significantly helped us improve the clarity and rigor of our paper. If our rebuttal has addressed your concerns satisfactorily, we would be honored if you could consider reflecting this in your final score.

---

> ### Author Response · Authors · 2025-11-28
>
> Thank you for your thoughtful and constructive feedback on our work. We have carefully addressed your comments and revised the work and manuscript accordingly. *As the discussion period nears its end, we would greatly appreciate any additional questions or points of clarification.* If our responses have satisfactorily addressed your concerns, we kindly ask you to consider reflecting this in your score.
>
> Thanks again for your time and expertise.

---

### Note · Authors · 2025-12-25

I have read and agree with the venue's withdrawal policy on behalf of myself and my co-authors.